# SIMPLIFIED STATE SPACE LAYERS FOR SEQUENCE MODELING

**Jimmy T.H. Smith**[*, 1, 2], **Andrew Warrington**[*, 2, 3], **Scott W. Linderman**[2, 3]
[*]Equal contribution.
[1]Institute for Computational and Mathematical Engineering, Stanford University.
[2]Wu Tsai Neurosciences Institute, Stanford University.
[3]Department of Statistics, Stanford University.
`{jsmith14,awarring,scott.linderman}@stanford.edu.`

## ABSTRACT

Models using *structured state space sequence* (S4) layers have achieved state-of-the-art performance on long-range sequence modeling tasks. An S4 layer combines linear state space models (SSMs), the HiPPO framework, and deep learning to achieve high performance. We build on the design of the S4 layer and introduce a new state space layer, the *S5 layer*. Whereas an S4 layer uses many independent single-input, single-output SSMs, the S5 layer uses one multi-input, multi-output SSM. We establish a connection between S5 and S4, and use this to develop the initialization and parameterization used by the S5 model. The result is a state space layer that can leverage efficient and widely implemented parallel scans, allowing S5 to match the computational efficiency of S4, while also achieving state-of-the-art performance on several long-range sequence modeling tasks. S5 averages $87.4\%$ on the long range arena benchmark, and $98.5\%$ on the most difficult Path-X task.

## 1 INTRODUCTION

Efficiently modeling long sequences is a challenging problem in machine learning. Information crucial to solving tasks may be encoded jointly between observations that are thousands of timesteps apart. Specialized variants of recurrent neural networks (RNNs) (Arjovsky et al., 2016; Erichson et al., 2021; Rusch & Mishra, 2021; Chang et al., 2019), convolutional neural networks (CNNs) (Bai et al., 2018; Oord et al., 2016; Romero et al., 2022b), and transformers (Vaswani et al., 2017) have been developed to try to address this problem. In particular, many efficient transformer methods have been introduced (Choromanski et al., 2021; Katharopoulos et al., 2020; Kitaev et al., 2020; Beltagy et al., 2020; Gupta & Berant, 2020; Wang et al., 2020) to address the standard transformer's quadratic complexity in the sequence length. However, these more efficient transformers still perform poorly on very long-range sequence tasks (Tay et al., 2021).

Gu et al. (2021a) presented an alternative approach using *structured state space sequence* (S4) layers. An S4 layer defines a nonlinear sequence-to-sequence transformation via a bank of many independent single-input, single-output (SISO) linear state space models (SSMs) (Gu et al., 2021b), coupled together with nonlinear mixing layers. Each SSM leverages the HiPPO framework (Gu et al., 2020a) by initializing with specially constructed state matrices. Since the SSMs are linear, each layer can be equivalently implemented as a convolution, which can then be applied efficiently by parallelizing across the sequence length. Multiple S4 layers can be stacked to create a deep sequence model. Such models have achieved significant improvements over previous methods, including on the *long range arena* (LRA) (Tay et al., 2021) benchmarks specifically designed to stress test long-range sequence models. Extensions have shown good performance on raw audio generation (Goel et al., 2022) and classification of long movie clips (Islam & Bertasius, 2022).

We introduce a new state space layer that builds on the S4 layer, the *S5* layer, illustrated in Figure 1. S5 streamlines the S4 layer in two main ways. First, S5 uses one multi-input, multi-output (MIMO) SSM in place of the bank of many independent SISO SSMs in S4. Second, S5 uses an efficient and widely implemented parallel scan. This removes the need for the convolutional and frequency-domain approach used by S4, which requires a non-trivial computation of the convolution kernel.

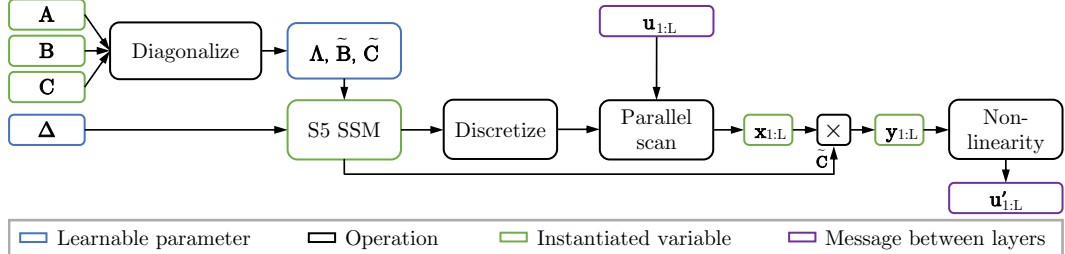

Figure 1: The computational components of an S5 layer for offline application to a sequence. The S5 layer uses a parallel scan on a diagonalized linear SSM to compute the SSM outputs $\mathbf{y}_{1:L} \in \mathbb{R}^{L \times H}$. A nonlinear activation function is applied to the SSM outputs to produce the layer outputs. A similar diagram for S4 is included in Appendix B.

The resulting state space layer has the same computational complexity as S4, but operates purely recurrently and in the time domain.

We then establish a mathematical relationship between S4 and S5. This connection allows us to inherit the HiPPO initialization schemes that are key to the success of S4. Unfortunately, the specific HiPPO matrix that S4 uses for initialization cannot be diagonalized in a numerically stable manner for use in S5. However, in line with recent work on the *DSS* (Gupta et al., 2022) and *S4D* (Gu et al., 2022) layers, we found that a diagonal approximation to the HiPPO matrix achieves comparable performance. We extend a result from Gu et al. (2022) to the MIMO setting, which justifies the diagonal approximation for use in S5. We leverage the mathematical relationship between S4 and S5 to inform several other aspects of parameterization and initialization, and we perform thorough ablation studies to explore these design choices.

The final S5 layer has many desirable properties. It is straightforward to implement (see Appendix A),[1] enjoys linear complexity in the sequence length, and can efficiently handle time-varying SSMs and irregularly sampled observations (which is intractable with the convolution implementation of S4). S5 achieves state-of-the-art performance on a variety of long-range sequence modeling tasks, with an LRA average of $87.4\%$, and $98.5\%$ accuracy on the most difficult Path-X task.

## 2 BACKGROUND

We provide the necessary background in this section prior to introducing the S5 layer in Section 3.

### 2.1 LINEAR STATE SPACE MODELS

Continuous-time linear SSMs are the core component of both the S4 layer and the S5 layer. Given an input signal $\mathbf{u}(t) \in \mathbb{R}^U$, a latent state $\mathbf{x}(t) \in \mathbb{R}^P$ and an output signal $\mathbf{y}(t) \in \mathbb{R}^M$, a linear continuous-time SSM is defined by the differential equation:

$$\frac{d\mathbf{x}(t)}{dt} = \mathbf{A}\mathbf{x}(t) + \mathbf{B}\mathbf{u}(t), \qquad \mathbf{y}(t) = \mathbf{C}\mathbf{x}(t) + \mathbf{D}\mathbf{u}(t), \tag{1}$$

and is parameterized by a state matrix $\mathbf{A} \in \mathbb{R}^{P \times P}$, an input matrix $\mathbf{B} \in \mathbb{R}^{P \times U}$, an output matrix $\mathbf{C} \in \mathbb{R}^{M \times P}$ and a feedthrough matrix $\mathbf{D} \in \mathbb{R}^{M \times U}$. For a constant step size, $\Delta$, the SSM can be *discretized* using, e.g. Euler, bilinear or zero-order hold (ZOH) methods to define the linear recurrence

$$\mathbf{x}_k = \overline{\mathbf{A}}\mathbf{x}_{k-1} + \overline{\mathbf{B}}\mathbf{u}_k, \qquad \mathbf{y}_k = \overline{\mathbf{C}}\mathbf{x}_k + \overline{\mathbf{D}}\mathbf{u}_k, \tag{2}$$

where the discrete-time parameters are each a function, specified by the discretization method, of the continuous-time parameters. See Iserles (2009) for more information on discretization methods.

### 2.2 PARALLELIZING LINEAR STATE SPACE MODELS WITH SCANS

We use parallel scans to efficiently compute the states of a discretized linear SSM. Given a binary associative operator $\bullet$ (i.e. $(a \bullet b) \bullet c = a \bullet (b \bullet c)$) and a sequence of $L$ elements $[a_1, a_2, ..., a_L]$, the

---

[1]The full S5 implementation is available at: https://github.com/lindermanlab/S5.

scan operation (sometimes referred to as *all-prefix-sum*) returns the sequence

$$[a_1, \ (a_1 \bullet a_2), \ ..., \ (a_1 \bullet a_2 \bullet ... \bullet a_L)]. \tag{3}$$

Computing a length $L$ linear recurrence of a discretized SSM, $\mathbf{x}_k = \overline{\mathbf{A}}\mathbf{x}_{k-1} + \overline{\mathbf{B}}\mathbf{u}_k$ as in (2), is a specific example of a scan operation. As discussed in Section 1.4 of Blelloch (1990), parallelizing the linear recurrence of the latent transitions in the discretized SSM above can be computed in a parallel time of $\mathcal{O}(T_\odot \log L)$, assuming $L$ processors, where $T_\odot$ represents the cost of matrix-matrix multiplication. For a general matrix $\overline{\mathbf{A}} \in \mathbb{R}^{P \times P}$, $T_\odot$ is $\mathcal{O}(P^3)$. This can be prohibitively expensive in deep learning settings. However, if $\overline{\mathbf{A}}$ is a diagonal matrix, the parallel time becomes $\mathcal{O}(P \log L)$ with $L$ processors and only requires $\mathcal{O}(PL)$ space. Finally, we note that efficient parallel scans are implemented in a work-efficient manner, thus the total computational cost of the parallel scan with a diagonal matrix is $\mathcal{O}(PL)$ operations. See Appendix H for more information on parallel scans.

## 2.3 S4: STRUCTURED STATE SPACE SEQUENCE LAYERS

The S4 layer (Gu et al., 2021a) defines a nonlinear sequence-to-sequence transformation, mapping from an input sequence $\mathbf{u}_{1:L} \in \mathbb{R}^{L \times H}$ to an output sequence $\mathbf{u}'_{1:L} \in \mathbb{R}^{L \times H}$. An S4 layer contains a bank of $H$ independent single-input, single-output (SISO) SSMs with $N$-dimensional states. Each *S4 SSM* is applied to one dimension of the input sequence. This results in an independent linear transformation from each input channel to each preactivation channel. A nonlinear activation function is then applied to the preactivations. Finally, a position-wise linear *mixing* layer is applied to combine the independent features and produce the output sequence $\mathbf{u}'_{1:L}$. Figure 4a in the appendix illustrates the view of the S4 layer as a bank of independent SSMs. Figure 2a shows an alternative view of S4 as one large SSM with state size $HN$ and block-diagonal state, input and output matrices.

Each S4 SSM leverages the HiPPO framework for online function approximation (Gu et al., 2020a) by initializing the state matrices with a HiPPO matrix (most often the HiPPO-LegS matrix). This was demonstrated empirically to lead to strong performance (Gu et al., 2021b;a), and can be shown as approximating long-range dependencies with respect to an infinitely long, exponentially-decaying measure (Gu et al., 2023). While the HiPPO-LegS matrix is not stably diagonalizable (Gu et al., 2021a), it can be represented as a normal plus low-rank (NPLR) matrix. The normal component, referred to as HiPPO-N and denoted $\mathbf{A}_{\text{LegS}}^{\text{Normal}}$, can be diagonalized. Thus, the HiPPO-LegS can be conjugated into a diagonal plus low-rank (DPLR) form, which S4 then utilizes to derive an efficient form of the convolution kernel. This motivates S4's DPLR parameterization.

Efficiently applying the S4 layer requires two separate implementations depending on context: a recurrent mode and a convolution mode. For online generation, the SSM is iterated recurrently, much like other RNNs. However, when the entire sequence is available and the observations are evenly spaced, a more efficient convolution mode is used. This takes advantage of the ability to represent the linear recurrence as a one-dimensional convolution between the inputs and a convolution kernel for each of the SSMs. Fast Fourier transforms (FFTs) can then be applied to efficiently parallelize this application. Figure 4a in the appendix illustrates the convolution approach of the S4 layer for offline processing. We note that while parallel scans could, in principle, allow a recurrent approach to be used in offline scenarios, applying the parallel scan to all $H$ of the $N$-dimensional SSMs would in general be much more expensive than the convolution approach S4 actually uses.

The trainable parameters of each S4 layer are the $H$ independent copies of the learnable SSM parameters and the $\mathcal{O}(H^2)$ parameters of the mixing layer. For each of the $h \in \{1, ..., H\}$ S4 SSMs, given a scalar input signal $u^{(h)}(t) \in \mathbb{R}$, an S4 SSM uses an input matrix $\mathbf{B}^{(h)} \in \mathbb{C}^{N \times 1}$, a DPLR parameterized transition matrix $\mathbf{A}^{(h)} \in \mathbb{C}^{N \times N}$, an output matrix $\mathbf{C}^{(h)} \in \mathbb{C}^{1 \times N}$, and feedthrough matrix $\mathbf{D}^{(h)} \in \mathbb{R}^{1 \times 1}$, to produce a signal $y^{(h)}(t) \in \mathbb{R}$. To apply the S4 SSMs to discrete sequences, each continuous-time SSM is discretized using a constant timescale parameter $\Delta^{(h)} \in \mathbb{R}_+$. The learnable parameters of each SSM are the timescale parameter $\Delta^{(h)} \in \mathbb{R}_+$, the continuous-time parameters $\mathbf{B}^{(h)}$, $\mathbf{C}^{(h)}$, $\mathbf{D}^{(h)}$, and the DPLR matrix, parameterized by vectors $\mathbf{\Lambda}^{(h)} \in \mathbb{C}^N$ and $\mathbf{p}^{(h)}, \mathbf{q}^{(h)} \in \mathbb{C}^N$ representing the diagonal matrix and low-rank terms respectively. For notational compactness we denote the concatenation of the $H$ S4 SSM states at discrete time index $k$ as $\mathbf{x}_k^{(1:H)} = \left[ (\mathbf{x}_k^{(1)})^\top, \ldots, (\mathbf{x}_k^{(H)})^\top \right]^\top$, and the $H$ SSM outputs as $\mathbf{y}_k = \left[ \mathbf{y}_k^{(1)}, \ldots, \mathbf{y}_k^{(H)} \right]^\top$.

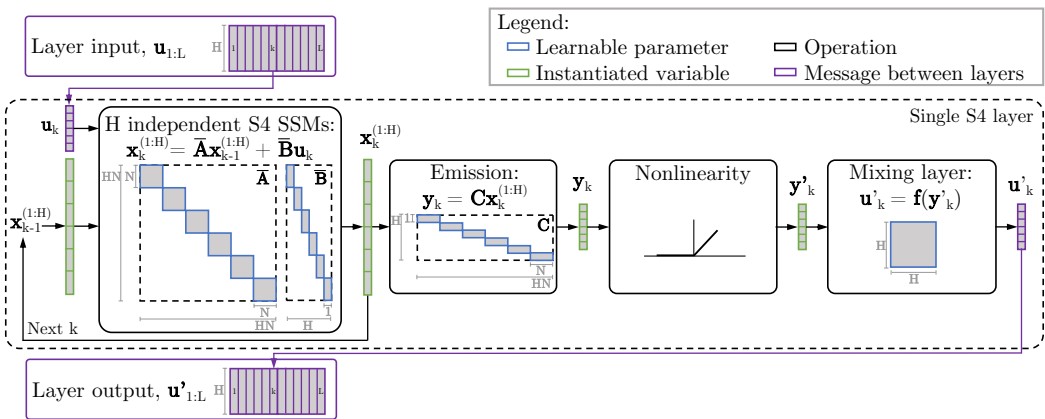

(a) Internal structure of a single S4 layer (Gu et al., 2021a) when viewed as a block-diagonal system.

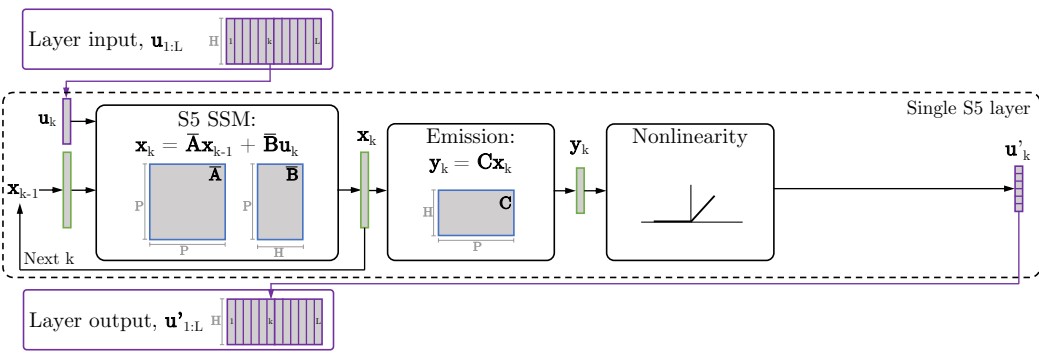

(b) Internal structure of a single S5 layer.

Figure 2: Schematic of the internal structure of a discretized S4 layer (Gu et al., 2021a) (top) and S5 layer (bottom). Note $\mathbf{D}$ is omitted for simplicity. We view an S4 layer as a single block-diagonal SSM with a latent state of size $HN$, followed by a nonlinearity and mixing layer to mix the independent features. (b) In contrast, the S5 layer uses a dense, MIMO linear SSM with latent size $P \ll HN$.

# 3   THE S5 LAYER

In this section we present the S5 layer. We describe its structure, parameterization and computation, particularly focusing on how each of these differ from S4.

## 3.1   S5 STRUCTURE: FROM SISO TO MIMO

The S5 layer replaces the bank of SISO SSMs (or large block-diagonal system) in S4 with a multi-input, multi-output (MIMO) SSM, as in (1), with a latent state size $P$, and input and output dimension $H$. The discretized version of this MIMO SSM can be applied to a vector-valued input sequence $\mathbf{u}_{1:L} \in \mathbb{R}^{L \times H}$, to produce a vector-valued sequence of *SSM outputs* (or preactivations) $\mathbf{y}_{1:L} \in \mathbb{R}^{L \times H}$, using latent states $\mathbf{x}_k \in \mathbb{R}^P$. A nonlinear activation function is then applied to produce a sequence of layer outputs $\mathbf{u}'_{1:L} \in \mathbb{R}^{L \times H}$. See Figure 2b for an illustration. Unlike S4, we do not require an additional position-wise linear layer, since these features are already mixed. We note here that compared to the $HN$ latent size of the block-diagonal SSM in the S4 layer, S5's latent size $P$ can be significantly smaller, allowing for the use of efficient parallel scans, as we discuss in Section 3.3.

## 3.2   S5 PARAMETERIZATION: DIAGONALIZED DYNAMICS

The parameterization of the S5 layer's MIMO SSM is motivated by the desire to use efficient parallel scans. As discussed in Section 2.2, a diagonal state matrix is required to efficiently compute the linear recurrence using a parallel scan. Thus, we diagonalize the system, writing the continuous-time state matrix as $\mathbf{A} = \mathbf{V}\mathbf{\Lambda}\mathbf{V}^{-1}$, where $\mathbf{\Lambda} \in \mathbb{C}^{P \times P}$ denotes the diagonal matrix containing the eigenvalues

and $\mathbf{V} \in \mathbb{C}^{P \times P}$ corresponds to the eigenvectors. Therefore, we can diagonalize the continuous-time latent dynamics from (1) as

$$\frac{\mathrm{d}\mathbf{V}^{-1}\mathbf{x}(t)}{\mathrm{d}t} = \mathbf{\Lambda}\mathbf{V}^{-1}\mathbf{x}(t) + \mathbf{V}^{-1}\mathbf{B}\mathbf{u}(t). \tag{4}$$

Defining $\tilde{\mathbf{x}}(t) = \mathbf{V}^{-1}\mathbf{x}(t)$, $\tilde{\mathbf{B}} = \mathbf{V}^{-1}\mathbf{B}$, and $\tilde{\mathbf{C}} = \mathbf{C}\mathbf{V}$ gives a reparameterized system,

$$\frac{\mathrm{d}\tilde{\mathbf{x}}(t)}{\mathrm{d}t} = \mathbf{\Lambda}\tilde{\mathbf{x}}(t) + \tilde{\mathbf{B}}\mathbf{u}(t), \qquad \mathbf{y}(t) = \tilde{\mathbf{C}}\tilde{\mathbf{x}}(t) + \mathbf{D}\mathbf{u}(t). \tag{5}$$

This is a linear SSM with a diagonal state matrix. This diagonalized system can be discretized with a timescale parameter $\Delta \in \mathbb{R}_+$ using the ZOH method to give another diagonalized system with parameters

$$\overline{\mathbf{\Lambda}} = e^{\mathbf{\Lambda}\Delta}, \quad \overline{\mathbf{B}} = \mathbf{\Lambda}^{-1}(\overline{\mathbf{\Lambda}} - \mathbf{I})\tilde{\mathbf{B}}, \quad \overline{\mathbf{C}} = \tilde{\mathbf{C}}, \quad \overline{\mathbf{D}} = \mathbf{D}. \tag{6}$$

In practice, we use a vector of learnable timescale parameters $\mathbf{\Delta} \in \mathbb{R}^P$ (see Section 4.3) and restrict the feedthrough matrix $\mathbf{D}$ to be diagonal. The S5 layer therefore has the learnable parameters: $\tilde{\mathbf{B}} \in \mathbb{C}^{P \times H}$, $\tilde{\mathbf{C}} \in \mathbb{C}^{H \times P}$, $\mathrm{diag}(\mathbf{D}) \in \mathbb{R}^H$, $\mathrm{diag}(\mathbf{\Lambda}) \in \mathbb{C}^P$, and $\mathbf{\Delta} \in \mathbb{R}^P$.

**Initialization**   Prior work showed that the performance of deep state space models are sensitive to the initialization of the state matrix (Gu et al., 2021b;a). We discussed in Section 2.2 that state matrices must be diagonal for efficient application of parallel scans. We also discussed in Section 2.3 that the HiPPO-LegS matrix cannot be diagonalized stably, but that the HiPPO-N matrix can be. In Section 4 we connect the dynamics of S5 to S4 to suggest why initializing with HiPPO-like matrices may also work well in the MIMO setting. We support this empirically, finding that diagonalizing the HiPPO-N matrix leads to good performance, and perform ablations in Appendix E to compare to other initializations. We note that DSS (Gupta et al., 2022) and S4D (Gu et al., 2022) layers also found strong performance in the SISO setting by using a diagonalization of the HiPPO-N matrix.

**Conjugate Symmetry**   The complex eigenvalues of a diagonalizable matrix with real entries always occur in conjugate pairs. We enforce this conjugate symmetry by using half the number of eigenvalues and latent states. This ensures real outputs and reduces the runtime and memory usage of the parallel scan by a factor of two. This idea is also discussed in Gu et al. (2022).

### 3.3   S5 Computation: Fully Recurrent

Compared to the large $HN$ effective latent size of the block-diagonal S4 layer, the smaller latent dimension of the S5 layer ($P$) allows the use of efficient parallel scans when the entire sequence is available. The S5 layer can therefore be efficiently used as a recurrence in the time domain for both online generation and offline processing. Parallel scans and the continuous-time parameterization also allow for efficient handling of irregularly sampled time series and other time-varying SSMs, by simply supplying a different $\overline{\mathbf{A}}_k$ matrix at each step. We leverage this feature and apply S5 to irregularly sampled data in Section 6.3. In contrast, the convolution of the S4 layer requires a time invariant system and regularly spaced observations.

### 3.4   Matching the Computational Efficiency of S4 and S5

A key design desiderata for S5 was matching the computational complexity of S4 for both online generation and offline recurrence. The following proposition guarantees that their complexities are of the same order if S5's latent size $P = \mathcal{O}(H)$.

**Proposition 1.** *Given an S4 layer with $H$ input/output features, an S5 layer with $H$ input/output features and a latent size $P = \mathcal{O}(H)$ has the same order of magnitude complexity as an S4 layer in terms of both runtime and memory usage.*

*Proof.*   See Appendix C.1.                                                                                    □

We also support this proposition with empirical comparisons in Appendix C.2.

# 4 RELATIONSHIP BETWEEN S4 AND S5

We now establish a relationship between the dynamics of S5 and S4. In Section 4.1 we show that, under certain conditions, the outputs of the S5 SSM can be *interpreted* as a projection of the latent states computed by a particular S4 system. This interpretation motivates using HiPPO initializations for S5, which we discuss in more detail in Section 4.2. In Section 4.3 we discuss how the conditions required to relate the dynamics further motivate initialization and parameterization choices.

## 4.1 DIFFERENT OUTPUT PROJECTIONS OF EQUIVALENT DYNAMICS

We compare the dynamics of S4 and S5 under some simplifying assumptions:

**Assumption 1.** *We consider only $H$-dimensional to $H$-dimensional sequence maps.*

**Assumption 2.** *We assume the state matrix of each S4 SSM is identical, $\mathbf{A}^{(h)} = \mathbf{A} \in \mathbb{C}^{N \times N}$.*

**Assumption 3.** *We assume the timescales of each S4 SSM are identical, $\Delta^{(h)} = \Delta \in \mathbb{R}_+$*

**Assumption 4.** *We assume that the same state matrix $\mathbf{A}$ is used in S5 as in S4 (also cf. Assumption 2). Note this also specifies the S5 latent size $P = N$. We also assume the S5 input matrix is the horizontal concatenation of the column input vectors used by S4: $\mathbf{B} \triangleq \left[ \mathbf{B}^{(1)} \mid \ldots \mid \mathbf{B}^{(H)} \right]$.*

We will discuss relaxing these assumptions shortly, but under these conditions it is straightforward to derive a relationship between the dynamics of S4 and S5:

**Proposition 2.** *Consider an S5 layer, with state matrix $\mathbf{A}$, input matrix $\mathbf{B}$ and some output matrix $\mathbf{C}$ (cf. Assumption 1); and an S4 layer, where each of the $H$ S4 SSMs has state matrix $\mathbf{A}$ (cf. Assumption 2, 4) and input vector $\mathbf{B}^{(h)}$ (cf. Assumption 4). If the S4 and S5 layers are discretized with the same timescales (cf. Assumption 3), then the S5 SSM produces outputs, $\mathbf{y}_k$, equivalent to a linear combination of the latent states of the $H$ S4 SSMs, $\mathbf{y}_k = \mathbf{C}^{\mathrm{equiv}} \mathbf{x}_k^{(1:H)}$, where $\mathbf{C}^{\mathrm{equiv}} = [\, \mathbf{C} \; \cdots \; \mathbf{C} \,]$.*

*Proof.* See Appendix D.2. □

Importantly, the S5 SSM outputs are *not* equal to the outputs of the block-diagonal S4 SSM. Instead they are equivalent to the outputs of the block-diagonal S4 SSM with modified output matrix $\mathbf{C}^{\mathrm{equiv}}$. Under the assumptions, however, the underlying state dynamics *are* equivalent. Recalling that initializing the S4 dynamics with HiPPO was key to performance (Gu et al., 2021a), the relationship established in Proposition 2 motivates using HiPPO initializations for S5, as we now discuss.

## 4.2 DIAGONALIZABLE INITIALIZATION

Ideally, given the interpretation above, we would initialize S5 with the exact HiPPO-LegS matrix. Unfortunately, as discussed in Section 2.3, this matrix is not stably diagonalizable, as is required for the efficient parallel scans used for S5. However, Gupta et al. (2022) and Gu et al. (2022) showed empirically that removing the low rank terms and initializing with the diagonalized HiPPO-N matrix still performed well. Gu et al. (2022) offered a theoretical justification for the use of this normal approximation for single-input systems: in the limit of infinite state dimension, the linear ODE with HiPPO-N state matrix produces the same dynamics as an ODE with the HiPPO-LegS matrix. Using linearity, it is straightforward to extend this result to the multi-input system that S5 uses:

**Corollary 1** (Extension of Theorem 3 in Gu et al. (2022))**.** *Consider $\mathbf{A}_{\mathrm{LegS}} \in \mathbb{R}^{N \times N}$, $\mathbf{A}_{\mathrm{LegS}}^{\mathrm{Normal}} \in \mathbb{R}^{N \times N}$, $\mathbf{B}_{\mathrm{LegS}} \in \mathbb{R}^{N \times H}, \mathbf{P}_{\mathrm{LegS}} \in \mathbb{R}^N$ as defined in Appendix B.1.1. Given vector-valued inputs $\mathbf{u}(t) \in \mathbb{R}^H$, the ordinary differential equation $\dfrac{\mathrm{d}\mathbf{x}'(t)}{\mathrm{d}t} = \mathbf{A}_{\mathrm{LegS}}^{\mathrm{Normal}} \mathbf{x}'(t) + \frac{1}{2} \mathbf{B}_{\mathrm{LegS}} \mathbf{u}(t)$ converges to $\dfrac{\mathrm{d}\mathbf{x}(t)}{\mathrm{d}t} = \mathbf{A}_{\mathrm{LegS}} \mathbf{x}(t) + \mathbf{B}_{\mathrm{LegS}} \mathbf{u}(t)$ as $N \to \infty$.*

We include a simple proof of this extension in Appendix D.3. This extension motivates the use of HiPPO-N to initialize S5's MIMO SSM. Note that S4D (the diagonal extension of S4) uses the same HiPPO-N matrix. Thus, when under the assumptions in Proposition 2, an S5 SSM in fact produces outputs that are equivalent to a linear combination of the latent states produced by S4D's SSMs. Our empirical results in Section 6 suggest that S5 initialized with the HiPPO-N matrix performs just as well as S4 initialized with the HiPPO-LegS matrix.

### 4.3 RELAXING THE ASSUMPTIONS

We now revisit the assumptions required for Proposition 2, since they only relate a constrained version of S5 to a constrained version of S4. Regarding Assumption 2, Gu et al. (2021a) report that S4 models with tied state matrices can still perform well, though allowing different state matrices often yields higher performance. Likewise, requiring a single scalar timescale across all of the S4 SSMs, per Assumption 3, is restrictive. S4 typically learns different timescale parameters for each SSM (Gu et al., 2023) to capture different timescales in the data. To relax these assumptions, note that Assumption 4 constrains S5 to have dimension $P = N$, and $N$ is typically much smaller than the dimensionality of the inputs, $H$. Proposition 1 established that S5 can match S4's complexity with $P = \mathcal{O}(H)$. By allowing for larger latent state sizes, Assumptions 2 and 3 can be relaxed, as discussed in Appendix D.4. We also discuss how this relaxation motivates a block-diagonal initialization with HiPPO-N matrices on the diagonal. Finally, to further relax the tied timescale assumptions, we note that in practice, we find improved performance by learning $P$ different timescales (one per state). See Appendix D.5 for further discussion of this empirical finding and the ablations in Appendix E.1.

## 5 RELATED WORK

S5 is most directly related to S4 and its other extensions, which we have discussed thoroughly. However, there is prior literature that uses similar ideas to those developed here. For example, prior work studied approximating nonlinear RNNs with stacks of linear RNNs connected by nonlinear layers, while also using parallel scans (Martin & Cundy, 2018). Martin & Cundy (2018) showed that several efficient RNNs, such as QRNNs (Bradbury et al., 2017) and SRUs (Lei et al., 2018), fall into a class of linear surrogate RNNs that can leverage parallel scans. Kaul (2020) also used parallel scans for an approach that approximates RNNs with stacks of discrete-time single-input, multi-output (SIMO) SSMs. However, S4 and S5 are the only methods to significantly outperform other comparable state-of-the-art nonlinear RNNs, transformers and convolution approaches. Our ablation study in Appendix E.2 suggests that this performance gain over prior attempts at parallelized linear RNNs is likely due to the continuous-time parameterization and the HiPPO initialization.

## 6 EXPERIMENTS

We now compare empirically the performance of the S5 layer to the S4 layer and other baseline methods. We use the S5 layer as a drop-in replacement for the S4 layer. The architecture consists of a linear input encoder, stacks of S5 layers, and a linear output decoder (Gu et al., 2021a). For all experiments we choose the S5 dimensions to ensure similar computational complexities as S4, following the conditions discussed in Section 3.3, as well as comparable parameter counts. The results we present show that the S5 layer matches the performance and efficiency of the S4 layer. We include in the appendix further ablations, baselines and runtime comparisons.

### 6.1 LONG RANGE ARENA

The long range arena (LRA) benchmark (Tay et al., 2021) is a suite of six sequence modeling tasks, with sequence lengths from 1,024 to over 16,000. The suite was specifically developed to benchmark the performance of architectures on long-range modeling tasks (see Appendix G for more details). Table 1 presents S5's LRA performance in comparison to other methods. S5 achieves the highest average score among methods that have linear complexity in sequence length (most notably S4, S4D, and the concurrent works: Liquid-S4 (Hasani et al., 2023) and Mega-chunk (Ma et al., 2023)). Most significantly, S5 achieves the highest score among all models (including Mega (Ma et al., 2023)) on the Path-X task, which has by far the longest sequence length of the tasks in the benchmark.

### 6.2 RAW SPEECH CLASSIFICATION

The Speech Commands dataset (Warden, 2018) contains high-fidelity sound recordings of different human readers reciting a word from a vocabulary of 35 words. The task is to classify which word was spoken. We show in Table 2 that S5 outperforms the baselines, outperforms previous S4 methods and performs similarly to the concurrent Liquid-S4 method (Hasani et al., 2023). As S4 and S5 methods

Table 1: Test accuracy on the LRA benchmark tasks (Tay et al., 2021). ✗ indicates the model did not exceed random guessing. We include an expanded table, Table 7, with full citations and error bars in the appendix. We follow the procedure reported in Gu et al. (2021a; 2022) and report means across three seeds for S4, S4D (as reported by Gu et al. (2021a; 2022)) and S5. Bold scores indicate highest performance, underlined scores indicate second placed performance. We also include the results for the concurrent methods *Liquid-S4* (Hasani et al., 2023) and *Mega* (Ma et al., 2023). Unlike S4 methods and S5, the best Mega model retains the transformer's $\mathcal{O}(L^2)$ complexity.

| Model (Input length) | ListOps (2,048) | Text (4,096) | Retrieval (4,000) | Image (1,024) | Pathfinder (1,024) | Path-X (16,384) | Avg. |
|---|---|---|---|---|---|---|---|
| Transformer | 36.37 | 64.27 | 57.46 | 42.44 | 71.40 | ✗ | 53.66 |
| Luna-256 | 37.25 | 64.57 | 79.29 | 47.38 | 77.72 | ✗ | 59.37 |
| H-Trans.-1D | 49.53 | 78.69 | 63.99 | 46.05 | 68.78 | ✗ | 61.41 |
| CCNN | 43.60 | 84.08 | ✗ | 88.90 | 91.51 | ✗ | 68.02 |
| Mega ($\mathcal{O}(L^2)$) | **63.14** | **90.43** | 91.25 | **90.44** | **96.01** | 97.98 | **88.21** |
| Mega-chunk ($\mathcal{O}(L)$) | 58.76 | 90.19 | 90.97 | 85.80 | 94.41 | 93.81 | 85.66 |
| S4D-LegS | 60.47 | 86.18 | 89.46 | 88.19 | 93.06 | 91.95 | 84.89 |
| S4-LegS | 59.60 | 86.82 | 90.90 | 88.65 | 94.20 | 96.35 | 86.09 |
| Liquid-S4 | 62.75 | 89.02 | 91.20 | 89.50 | 94.8 | 96.66 | 87.32 |
| **S5** | 62.15 | 89.31 | **91.40** | 88.00 | 95.33 | **98.58** | 87.46 |

Table 2: Test accuracy on 35-way Speech Commands classification task (Warden, 2018). We include an expanded table, Table 8, with error bars in the appendix. Training examples are one-second 16kHz audio waveforms. Last column indicates 0-shot testing at 8kHz (constructed by naive decimation). As in Gu et al. (2022), the mean across three random seeds is reported. Performance for the baselines InceptionNet through to S4D-Lin are reported from Gu et al. (2022).

| Model (Input length) | Parameters | 16kHz (16,000) | 8kHz (8,000) |
|---|---|---|---|
| InceptionNet (Nonaka & Seita, 2021) | 481K | 61.24 | 05.18 |
| ResNet-1 (Nonaka & Seita, 2021) | 216K | 77.86 | 08.74 |
| XResNet-50 (Nonaka & Seita, 2021) | 904K | 83.01 | 07.72 |
| ConvNet (Nonaka & Seita, 2021) | 26.2M | 95.51 | 07.26 |
| S4-LegS (Gu et al., 2021a) | 307K | 96.08 | 91.32 |
| S4D-LegS (Gu et al., 2022) | 306K | 95.83 | 91.08 |
| Liquid-S4 (Hasani et al., 2023) | 224K | **96.78** | 90.00 |
| **S5** | 280K | 96.52 | **94.53** |

are parameterized in continuous-time, these models can be applied to datasets with different sampling rates without the need for re-training, simply by globally re-scaling the timescale parameter $\Delta$ by the ratio between the new and old sampling rates. The result of applying the best S5 model *trained on 16kHz data*, to the speech data sampled (via decimation) at 8kHz, without any additional fine-tuning, is also presented in Table 2. S5 also improves this metric over the baseline methods.

## 6.3 VARIABLE OBSERVATION INTERVAL

The final application we study here highlights how S5 can naturally handle observations received at irregular intervals. S5 does so by supplying a different $\Delta_t$ value to the discretization at each step. We use the pendulum regression example presented by Becker et al. (2019) and Schirmer et al. (2022), illustrated in Figure 3. The input sequence is a sequence of $L = 50$ images, each $24 \times 24$ pixels in size, that has been corrupted with a correlated noise process and sampled at irregular intervals from a continuous trajectory of duration $T = 100$. The targets are the sine and cosine of the angle of the pendulum, which follows a nonlinear dynamical system. The velocity is unobserved. We match the

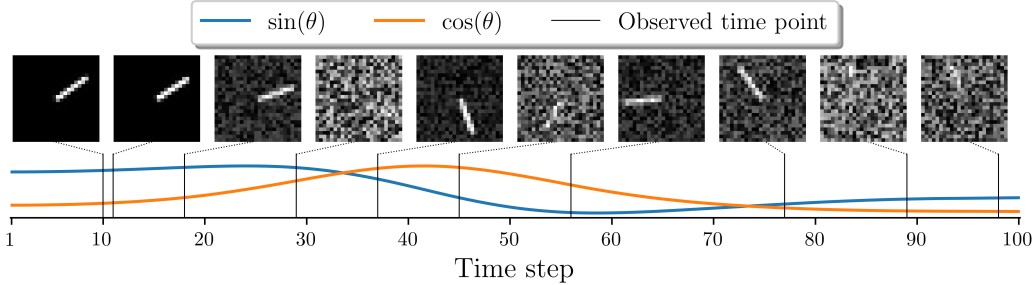

Figure 3: Illustration of the pendulum regression example. Shown in the top row are the images used as input at the time points indicated. Shown on the bottom are the values of $\sin(\theta_t)$ and $\cos(\theta_t)$, where $\theta_t$ is the angle of the pendulum at time $t$, that are used as the regression targets.

Table 3: Regression MSE $\times 10^{-3}$ (mean $\pm$ std) and relative application speed on the pendulum regression task on a held-out test set. Performance for the baselines, mTAND through to CRU, are reported from Schirmer et al. (2022). We include an expanded table, Table 9, and further details in the appendix. Results for *CRU (our run)* and *S5* are across twenty seeds.

| Model | Relative speed | Regression MSE ($\times 10^{-3}$) |
|---|---|---|
| mTAND (Shukla & Marlin, 2021) | 12.2× | 65.64 (4.05) |
| RKN (Becker et al., 2019) | 1.9× | 8.43 (0.61) |
| RKN-$\Delta_t$ (Becker et al., 2019) | 1.9× | 5.09 (0.40) |
| ODE-RNN (Rubanova et al., 2019) | 1.0× | 7.26 (0.41) |
| CRU (Schirmer et al., 2022) | 1.0× | 4.63 (1.07) |
| CRU (our run) | 1.0× | 3.94 (0.21) |
| **S5** | **86×** | **3.41** (0.27) |

architecture, parameter count and training procedure of Schirmer et al. (2022). Table 3 summarizes the results of this experiment. S5 outperforms CRU on the regression task, recovering a lower mean error. Furthermore, S5 is markedly faster than CRU on the same hardware.

## 6.4 PIXEL-LEVEL 1-D IMAGE CLASSIFICATION

Table 10 in Appendix F.4 shows results of S5 on other common benchmarks including sequential MNIST, permuted sequential MNIST and sequential CIFAR (color). We see that S5 broadly matches the performance of S4, and outperforms a range of state-of-the-art RNN-based methods.

## 7 CONCLUSION

We introduce the S5 layer for long-range sequence modeling. The S5 layer modifies the internal structure of the S4 layer, and replaces the frequency-domain approach used by S4 with a purely recurrent, time-domain approach leveraging parallel scans. S5 achieves high performance while retaining the computational efficiency of S4. S5 also provides further opportunities. For instance, unlike the convolutional S4 methods, the parallel scan unlocks the ability to efficiently and easily process time-varying SSMs whose parameters can vary with time. Section 6.3 illustrated an example of this for sequences sampled at variable sampling rates. The concurrently developed method, *Liquid-S4* (Hasani et al., 2023), uses an input-dependent bilinear dynamical system and highlights further opportunities for time-varying SSMs. The more general MIMO SSM design will also enable connections to be made with classical probabilistic state space modeling as well as more recent work on parallelizing filtering and smoothing operations (Särkkä & García-Fernández, 2020). More broadly, we hope the simplicity and generality of the S5 layer can expand the use of state space layers in deep sequence modeling and lead to new formulations and extensions.

ACKNOWLEDGEMENTS AND DISCLOSURE OF FUNDING

We thank Albert Gu for his thorough and insightful feedback. We also acknowledge The Annotated S4 Blog (Rush & Karamcheti, 2022) which inspired our JAX implementation. This work was supported by grants from the Simons Collaboration on the Global Brain (SCGB 697092), the NIH BRAIN Initiative (U19NS113201 and R01NS113119), and the Sloan Foundation. Some of the computation for this work was made possible by Stanford Data Science Microsoft Education Azure cloud credits.

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

APPENDIX FOR: SIMPLIFIED STATE SPACE LAYERS FOR SEQUENCE MODELING

CONTENTS:

# A  JAX IMPLEMENTATION OF S5 LAYER

```
1  import jax
2  import jax.numpy as np
3  parallel_scan = jax.lax.associative_scan
4
5  def discretize(Lambda, B_tilde, Delta):
6      """ Discretize a diagonalized, continuous-time linear SSM
7          Args:
8              Lambda  (complex64): diagonal state matrix                (P,)
9              B_tilde (complex64): input matrix                         (P, H)
10             Delta   (float32):   discretization step sizes            (P,)
11         Returns:
12             discretized Lambda_bar (complex64), B_bar (complex64)     (P,), (P,H)"""
13     Identity = np.ones(Lambda.shape[0])
14     Lambda_bar = np.exp(Lambda * Delta)
15     B_bar = (1 / Lambda * (Lambda_bar - Identity))[..., None] * B_tilde
16     return Lambda_bar, B_bar
17
18 def binary_operator(element_i, element_j):
19     """ Binary operator for parallel scan of linear recurrence. Assumes a diagonal matrix A.
20         Args:
21             element_i: tuple containing A_i and Bu_i at position i     (P,), (P,)
22             element_j: tuple containing A_j and Bu_j at position j     (P,), (P,)
23         Returns:
24             new element ( A_out, Bu_out )    """
25     A_i, Bu_i = element_i
26     A_j, Bu_j = element_j
27     return A_j * A_i, A_j * Bu_i + Bu_j
28
29 def apply_ssm(Lambda_bar, B_bar, C_tilde, D, input_sequence):
30     """ Compute the LxH output of discretized SSM given an LxH input.
31         Args:
32             Lambda_bar (complex64): discretized diagonal state matrix     (P,)
33             B_bar      (complex64): discretized input matrix              (P, H)
34             C_tilde    (complex64): output matrix                         (H, P)
35             D          (float32):   feedthrough matrix                    (H,)
36             input_sequence (float32): input sequence of features         (L, H)
37         Returns:
38             ys (float32): the SSM outputs (S5 layer preactivations)       (L, H)    """
39     # Prepare elements required to initialize parallel scan
40     Lambda_elements = np.repeat(Lambda_bar[None, ...], input_sequence.shape[0], axis=0)
41     Bu_elements = jax.vmap(lambda u: B_bar @ u)(input_sequence)
42     elements = (Lambda_elements, Bu_elements)                              # (L, P), (L, P)
43
44     # Compute latent state sequence given input sequence using parallel scan
45     _, xs = parallel_scan(binary_operator, elements)                      # (L, P)
46
47     # Compute SSM output sequence
48     ys = jax.vmap(lambda x, u: (C_tilde @ x + D * u).real)(xs, input_sequence)
49     return ys
50
51 def apply_S5_layer(params, input_sequence):
52     """ Computes LxH output sequence of an S5 layer given LxH input sequence.
53         Args:
54             params: tuple of the continuous time SSM parameters
55             input_sequence: input sequence of features                    (L, H)
56         Returns:
57             The S5 layer output sequence                                  (L, H)    """
58     Lambda, B_tilde, C_tilde, D, log_Delta = params
59     Lambda_bar, B_bar = discretize(Lambda, B_tilde, np.exp(log_Delta))
60     preactivations = apply_ssm(Lambda_bar, B_bar, C_tilde, D, input_sequence)
61     return jax.nn.gelu(preactivations)
62
63 def batch_apply_S5_layer(params, input_sequences):
64     """ Computes BxLxH output sequence of an S5 layer given BxLxH input sequence.
65         Args:
66             params: tuple of the continuous time SSM parameters
67             input_sequences: batch of input feature sequences            (B, L ,H)
68         Returns:
69             Batch of S5 layer output sequences                           (B, L, H)   """
70     return jax.vmap(apply_S5_layer, in_axes=(None, 0))(params, input_sequences)
```

Listing 1: JAX implementation to apply a single S5 layer to a batch of input sequences.

## B  S5 LAYER DETAILS

### B.1  INITIALIZATION DETAILS

#### B.1.1  INITIALIZATION OF THE STATE MATRIX

Here we provide additional details to supplement the discussion of initialization in Section 3.2. Gu et al. (2023) explains the ability of S4 to capture long-range dependencies when using the HiPPO-LegS matrix via decomposing the input with respect to an infinitely long, exponentially decaying measure. The HiPPO-LegS matrix and corresponding SISO input vector are defined as

$$(\mathbf{A}_{\mathrm{LegS}})_{nk} = - \begin{cases} (2n+1)^{1/2}(2k+1)^{1/2}, & n > k \\ n+1, & n = k \\ 0, & n < k \end{cases} . \tag{7}$$

$$(\mathbf{b}_{\mathrm{LegS}})_n = (2n+1)^{\frac{1}{2}}. \tag{8}$$

Note that in Section 4.2, the input matrix $\mathbf{B}_{\mathrm{LegS}} \in \mathbb{R}^{N \times H}$ used in Corollary 1 is formed by concatenating $H$ copies of $\mathbf{b}_{\mathrm{LegS}} \in \mathbb{R}^N$.

Theorem 1 of Gu et al. (2021a) then shows that the HiPPO matrices in Gu et al. (2020a), $\mathbf{A}_{\mathrm{HiPPO}} \in \mathbb{R}^{N \times N}$ can be represented with a normal plus low-rank (NPLR) form consisting of a normal matrix, $\mathbf{A}_{\mathrm{HiPPO}}^{\mathrm{Normal}} = \mathbf{V}\mathbf{\Lambda}\mathbf{V}^* \in \mathbb{R}^{N \times N}$, and a low-rank term

$$\mathbf{A}_{\mathrm{HiPPO}} = \mathbf{A}_{\mathrm{HiPPO}}^{\mathrm{Normal}} - \mathbf{P}\mathbf{Q}^\top = \mathbf{V}\left(\mathbf{\Lambda} - (\mathbf{V}^*\mathbf{P})(\mathbf{V}^*\mathbf{Q})^*\right)\mathbf{V}^* \tag{9}$$

for unitary $\mathbf{V} \in \mathbb{C}^{N \times N}$, diagonal $\mathbf{\Lambda} \in \mathbb{C}^{N \times N}$, and low-rank factorization $\mathbf{P}, \mathbf{Q} \in \mathbb{R}^{N \times r}$. The right hand side of this equation shows HiPPO matrices can be conjugated into a diagonal plus low-rank (DPLR) form. The HiPPO-LegS matrix can therefore be written in terms of the normal HiPPO-N matrix and low-rank term $\mathbf{P}_{\mathrm{LegS}} \in \mathbb{R}^N$ (Goel et al., 2022) as

$$\mathbf{A}_{\mathrm{LegS}} = \mathbf{A}_{\mathrm{LegS}}^{\mathrm{Normal}} - \mathbf{P}_{\mathrm{Legs}}\mathbf{P}_{\mathrm{Legs}}^\top \tag{10}$$

where

$$\mathbf{A}_{\mathrm{LegS}_{nk}}^{\mathrm{Normal}} = - \begin{cases} (n+\frac{1}{2})^{1/2}(k+\frac{1}{2})^{1/2}, & n > k \\ \frac{1}{2}, & n = k \\ (n+\frac{1}{2})^{1/2}(k+\frac{1}{2})^{1/2}, & n < k \end{cases} . \tag{11}$$

$$\mathbf{P}_{\mathrm{Legs}_n} = (n+\frac{1}{2})^{\frac{1}{2}} \tag{12}$$

Our default is to set the S5 layer state matrix $\mathbf{A} = \mathbf{A}_{\mathrm{LegS}}^{\mathrm{Normal}} \in \mathbb{R}^{P \times P}$, and take the eigendecomposition of this matrix to recover the initial $\mathbf{\Lambda}$. We often find it beneficial to also use $\mathbf{V}$ and $\mathbf{V}^{-1} = \mathbf{V}^*$ to initialize $\tilde{\mathbf{B}}$ and $\tilde{\mathbf{C}}$, as described below.

As mentioned in Section 4.3, we also found that performance on many tasks benefited from initializing the S5 state matrix as block-diagonal, with each block on the diagonal equal to $\mathbf{A}_{\mathrm{LegS}}^{\mathrm{Normal}} \in \mathbb{R}^{R \times R}$, where $R$ here is less than the state dimension $P$, e.g. $R = \frac{P}{4}$ when 4 blocks are used on the diagonal. We then take the eigendecomposition of this matrix to initialize $\mathbf{\Lambda}$, as well as $\tilde{\mathbf{B}}$ and $\tilde{\mathbf{C}}$. We note that even in this case, $\tilde{\mathbf{B}}$ and $\tilde{\mathbf{C}}$ are still initialized in dense form and there is no constraint that requires $\mathbf{A}$ to remain block-diagonal during learning. In the hyperparameter table in Appendix G, the $J$ hyperparameter indicates the number of these HiPPO-N blocks used on the diagonal for initialization, where $J = 1$ indicates we used the default case of initializing with a single HiPPO-N matrix. We discuss the motivation for this block-diagonal initialization further in Appendix D.4.

#### B.1.2  INITIALIZATION OF INPUT, OUTPUT AND FEED-THROUGH MATRICES

In general, we explicitly initialize the input matrix $\tilde{\mathbf{B}}$ and output matrix $\tilde{\mathbf{C}}$ using the eigenvectors from the diagonalization of the initial state matrix. Specifically, we sample $\mathbf{B}$ and $\mathbf{C}$ and then initialize the (complex) learnable parameters $\tilde{\mathbf{B}}$ as $\tilde{\mathbf{B}} = \mathbf{V}^{-1}\mathbf{B}$ and $\tilde{\mathbf{C}} = \mathbf{C}\mathbf{V}$.

We initialize $\mathbf{D} \in \mathbb{R}^H$ by independently sampling each element from a standard normal distribution.

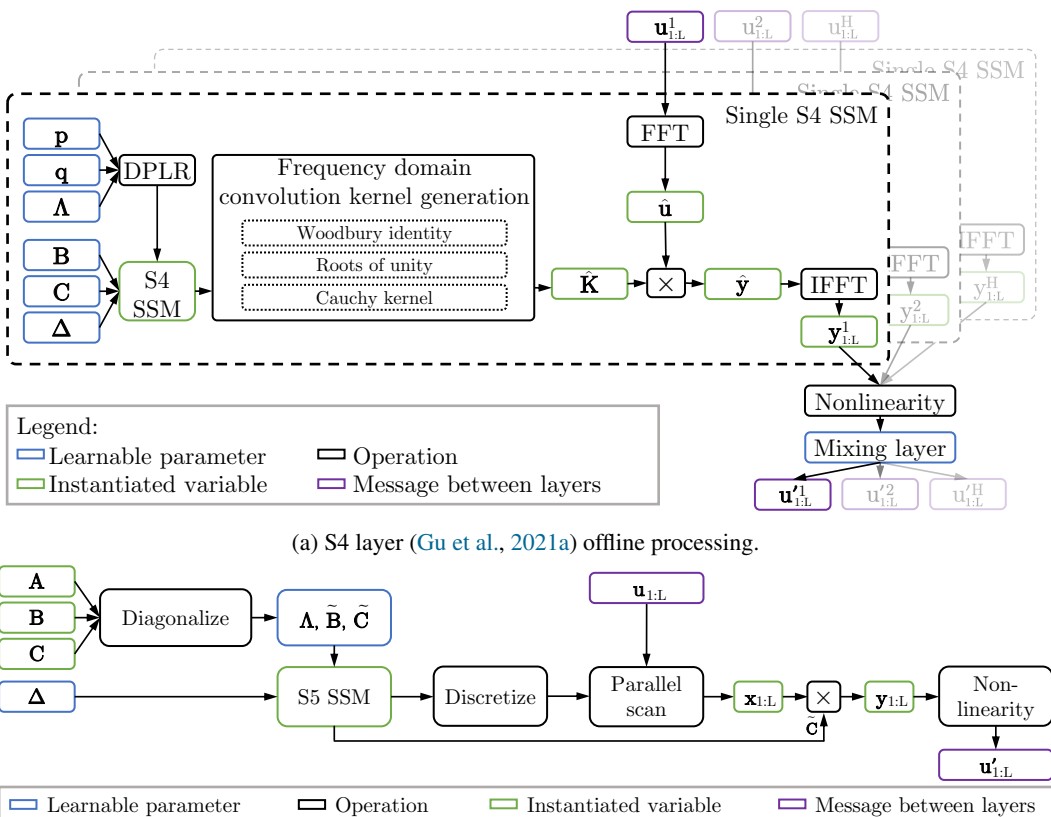

(a) S4 layer (Gu et al., 2021a) offline processing.

(b) S5 layer offline processing. Duplicated from the main text.

Figure 4: The computational components of the S4 layer (Gu et al., 2021a) (top) and the S5 layer (bottom) for offline application to a sequence. (a) The S4 layer applies an independent SSM to each dimension of the input sequence $\mathbf{u}_{1:L} \in \mathbb{R}^{L \times H}$. This requires a Cauchy kernel computation to compute the convolution kernel coefficients in the frequency domain. Convolutions are computed using FFTs to produce the independent SSM outputs $\mathbf{y}_{1:L} \in \mathbb{R}^{L \times H}$. A nonlinear activation function that includes a mixing layer is applied to the SSM outputs to produce the layer outputs. (b) (Reproduced from Figure 1) The S5 layer uses a parallel scan on a diagonalized linear SSM to compute the SSM outputs $\mathbf{y}_{1:L} \in \mathbb{R}^{L \times H}$. A nonlinear activation function is applied to the SSM outputs to produce the layer outputs.

### B.1.3 INITIALIZATION OF THE TIMESCALES

Prior work (Gupta et al., 2022; Gu et al., 2023) found the initialization of this timescale parameter to be important. This is studied in detail in Gu et al. (2023). We sample these parameters in line with S4 and sample each element of $\log \mathbf{\Delta} \in \mathbb{R}^P$ from a uniform distribution on the interval $[\log \delta_{\min}, \log \delta_{\max})$, where the default range is $\delta_{\min} = 0.001$ and $\delta_{\max} = 0.1$. The only exception is the Path-X experiment, where we initialize from $\delta_{\min} = 0.0001$ and $\delta_{\max} = 0.1$ to account for the longer timescales as discussed in Gu et al. (2023).

### B.2 COMPARISON OF S4 AND S5 COMPUTATIONAL ELEMENTS

In Figure 4 we illustrate a comparison of the computational details of the S4 and S5 layers for efficient, parallelized offline processing.

## C  COMPUTATIONAL EFFICIENCY OF S5

### C.1  THEORETICAL COMPUTATIONAL EFFICIENCY

**Proposition 1.** *Given an S4 layer with $H$ input/output features, an S5 layer with $H$ input/output features and a latent size $P = \mathcal{O}(H)$ has the same order of magnitude complexity as an S4 layer in terms of both runtime and memory usage.*

*Proof.* We first consider the case where the entire sequence is available and compare the S4 layer's convolution mode to the S5 layer's use of a parallel scan. We then consider the online generation case where each method operates recurrently.

**Parallelized offline processing**  We consider the application of both the S4 and S5 layer to a vector-valued sequence $\mathbf{u}_{1:L} \in \mathbb{R}^{L \times H}$. Note that there are multiple ways to compute the convolution kernel for S4/S4D and the exact computational complexity depends on the implementation (Gu et al., 2021a; 2022). However, the overall complexity of computing the S4 SSM outputs given the inputs is lower bounded by the FFTs used to convert into the frequency domain to apply the convolutions. Therefore, we can lower bound the cost of applying a single SISO S4 SSM to a scalar input sequence as $\mathcal{O}(L \log L)$ operations and $\mathcal{O}(L)$ space. The S4 layer then consists of $H$ different S4 SSMs, and therefore requires $\mathcal{O}(HL \log L)$ operations and $\mathcal{O}(HL)$ space. Finally, mixing the $H$ activations at each timestep requires $L$ independent matrix-vector multiplications. Thus, the S4 layer sequence-to-sequence transformation requires a total of $\mathcal{O}(H^2 L + HL \log L)$ operations when run in the efficient convolution mode. Given $H^2 L$ processors, these operations can be parallelized to a minimum of $\mathcal{O}(\log H + \log L)$ parallel time.

We now consider the S5 layer. As discussed in Section 2.2, the parallel scan requires $\mathcal{O}(PL)$ operations and $\mathcal{O}(PL)$ space to perform the linear recurrence that computes the discretized states $\mathbf{x}_{1:L} \in \mathbb{R}^{L \times P}$. In addition, $\mathcal{O}(PHL)$ operations are required for the independent matrix-vector multiplications that compute $\overline{\mathbf{B}}\mathbf{u}_{1:L} \in \mathbb{R}^{L \times P}$ and the SSM outputs $\tilde{\mathbf{C}}\mathbf{x}_{1:L} \in \mathbb{R}^{L \times H}$. Therefore, the S5 layer requires $\mathcal{O}(PHL + PL)$ operations. Given $PHL$ processors, these operations can be parallelized to a minimum of $\mathcal{O}(\log P + \log L)$ parallel time. Thus, we see that when the S5 state dimension $P = \mathcal{O}(H)$, the S5 layer requires $\mathcal{O}(H^2 L + HL)$ operations compared to the S4 layer's $\mathcal{O}(H^2 L + HL \log L)$ operations. Crucially, when $P = \mathcal{O}(H)$, S4 and S5 each have parallel complexity of $\mathcal{O}(\log H + \log L)$ (when $H^2 L$ processors are available). In addition, when $P = \mathcal{O}(H)$, the space complexity for the parallel scan is $\mathcal{O}(HL)$ which matches the space complexity of S4's FFTs. Both methods then also perform identical broadcasted matrix vector multiplications, and hence have the same space complexity.

**Online generation**  For online generation, both the S4 and S5 layers are run recurrently. The S4 layer requires $\mathcal{O}(H^2 + HN)$ operations per step due to its $\mathcal{O}(HN)$ DPLR-matrix-vector multiplication (Gu et al., 2021a) and the $\mathcal{O}(H^2)$ matrix-vector multiplication of its mixing layer. The S5 layer requires $\mathcal{O}(PH + P)$ operations per step due to its $\mathcal{O}(P)$ matrix-vector multiplication with its diagonal matrix and its $\mathcal{O}(PH)$ matrix-vector multiplications to compute $\overline{\mathbf{B}}\mathbf{u}_k \in \mathbb{R}^P$ and $\tilde{\mathbf{C}}\mathbf{x}_k \in \mathbb{R}^H$. Thus, we see the two approaches have the same per step complexity of $\mathcal{O}(H^2)$ when $P = \mathcal{O}(H)$ and the individual S4 SSM state sizes $N$ are $\mathcal{O}(H)$.

Thus, S4 and S5 have the same order computational complexity and memory requirements in both cases. □

### C.2  EMPIRICAL RUNTIME COMPARISON

Table 4 provides an empirical evaluation of the runtime performance, in terms of speed and memory, between S4, S4D and S5 across a range of sequence lengths from the LRA tasks. We compared the JAX implementation of S5 to a JAX implementation of S4 and S4D, based on the JAX implementation from Rush & Karamcheti (2022). For a fair comparison, we modified these existing JAX implementations of S4 and S4D to allow them both to enforce conjugate symmetry and use bidirectionality. For each task, models use bidirectionality and conjugate symmetry as reported in Gu et al. (2022). All models, except for the italicised S5 row, use the same input/output features $H$ and number of layers as reported in Gu et al. (2022). The S4 and S4D layers also use the same S4 SSM

Table 4: Benchmarking the runtime performance of S4, S4D and S5 on three LRA tasks of varied sequence lengths using parameterizations described in Section C.2. For speeds, $> 1\times$ indicates faster than the S4D baseline. For memory, $< 1\times$ indicates less memory was used than the S4D baseline. The fifth line of each metric shows the performance of the actual S5 model used for the LRA result in Table 1 for each task using the architecture reported in Table 11.

| Model | Architecture | Dataset (Input Length) | | |
|-------|-------------|-------------------|------|---------|
| | | ListOps (2,048) | Text (4,096) | Path-X (16,384) |
| | | Train step speed ($\uparrow$) | | |
| S4 | From Gu et al. (2021a) | 0.6 $\times$ | 0.4 $\times$ | 0.4 $\times$ |
| S4D | From Gu et al. (2022) | **1.0** $\times$ | 1.0 $\times$ | 1.0 $\times$ |
| S5 | (P=H) Matched to Gu et al. (2022) | 0.7 $\times$ | 0.9 $\times$ | 1.9 $\times$ |
| S5 | (P=N) Matched to Gu et al. (2022) | 0.9 $\times$ | **1.5** $\times$ | **2.9** $\times$ |
| *S5* | *As in Table 11* | *1.1* $\times$ | *1.0* $\times$ | *4.7* $\times$ |
| | | Evaluation step speed ($\uparrow$) | | |
| S4 | From Gu et al. (2021a) | 0.6 $\times$ | 0.5 $\times$ | 0.4 $\times$ |
| S4D | From Gu et al. (2022) | **1.0** $\times$ | 1.0 $\times$ | 1.0 $\times$ |
| S5 | (P=H) Matched to Gu et al. (2022) | 0.5 $\times$ | 0.9 $\times$ | 1.7 $\times$ |
| S5 | (P=N) Matched to Gu et al. (2022) | 0.6 $\times$ | **1.2** $\times$ | **1.8** $\times$ |
| *S5* | *As in Table 11* | *0.7* $\times$ | *1.0* $\times$ | *4.1* $\times$ |
| | | Memory allocation ($\downarrow$) | | |
| S4 | From Gu et al. (2021a) | 1.1 $\times$ | 1.2 $\times$ | 1.2 $\times$ |
| S4D | From Gu et al. (2022) | 1.0 $\times$ | 1.0 $\times$ | 1.0 $\times$ |
| S5 | (P=H) Matched to Gu et al. (2022) | 0.9 $\times$ | 1.2 $\times$ | 1.1 $\times$ |
| S5 | (P=N) Matched to Gu et al. (2022) | **0.7** $\times$ | **0.7** $\times$ | **0.7** $\times$ |
| *S5* | *As in Table 11* | *0.7* $\times$ | *1.0* $\times$ | *0.9* $\times$ |

latent size as reported in Gu et al. (2022). All methods used the same batch size and all comparisons were made using a 16GB NVIDIA V100 GPU. Note we observed the JAX S4D implementation to in general be faster than the JAX S4 implementation (possibly due to this specific S4 implementation's use of the naive Cauchy kernel computation (Gu et al., 2021a)). For this reason, we consider S4D as the baseline.

We consider three configurations of S5 for comparison. The first two configurations, corresponding to lines 3 and 4 for each metric in Table 4, show how the runtime metrics vary as S5's latent size is adjusted, with all other architecture choices equal to those of S4. In line 3 of each metric, we denote the S5 "Architecture" as "(P=H) Matched to Gu et al. (2022)" to indicate that this configuration of S5 sets the latent size equal to the number of input/output features, $P = H$. This line empirically supports the complexity argument presented in Appendix C.1. In line 4 of each metric, we denote the S5 "Architecture" as " (P=N) Matched to Gu et al. (2022)" to indicate that this configuration of S5 sets the latent size $P$ equal to the latent size $N$ S4 uses for each of its SISO SSMs. This line also corresponds to the constrained version of S5 that performs similarly to S4/S4D as presented in the ablation study in Table 5. The runtime results of both of these configurations supports the claim in Section 4.3 that the latent size of S5 can be increased while maintaining S4's computational efficiency.

Finally, we include a third configuration of S5, presented in the fifth line of each metric and *italicized*. This configuration of S5 uses the best architectural dimensions from Table 11 and was used for the corresponding LRA results in Table 1.

Importantly, the broad takeaway from this empirical study is that the runtime and memory usage of S5 and S4/S4D are broadly similar, as suggested by the complexity analysis in the main text.

# D RELATIONSHIP BETWEEN S4 AND S5

We now describe in more detail the connection between the S4 and S5 architectures. This connection allowed us to develop more performant architectures and extend theoretical results from existing work.

We break this analysis down into three parts:

1. In Section D.2 we prove Proposition 2. We exploit the linearity of the systems to identify that the latent states computed by the S5 SSM are equivalent to a linear combination of latent states computed by the $H$ SISO S4 SSMs, and that the outputs of the S5 SSM are a further linear transformation of these states. We then highlight how S4 and S5 effectively define different output matrices in the block-diagonal perspective shown in Figure 2.

2. In Section D.3 we provide a simple extension of the proof provided by Gu et al. (2022). The original proof shows that in the SISO case, in the limit of large $N$, the dynamics arising from a (non-diagonalizable) HiPPO-LegS matrix, are faithfully approximated by the (diagonalizable) normal component of the HiPPO-LegS matrix. We extend this proof to apply to the MIMO setting. This motivates initialization with the HiPPO-N matrix, which in-turn allows us to use parallel scans efficiently.

3. In Section D.4 we conclude by showing that, by judicious choice of initialization of the S5 state matrix, S5 can implement multiple independent S4 systems and relax the assumptions made. We also discuss the vector of timescale parameters, which we found to improve performance.

We note that many of these results follow straightforwardly from the linearity of the recurrence.

## D.1 ASSUMPTIONS

For these following sections we will use the following assumptions, until otherwise stated:

**Assumption 1.** *We consider only $H$-dimensional to $H$-dimensional sequence maps.*

**Assumption 2.** *We assume the state matrix of each S4 SSM is identical, $\mathbf{A}^{(h)} = \mathbf{A} \in \mathbb{C}^{N \times N}$.*

**Assumption 3.** *We assume the timescales of each S4 SSM are identical, $\Delta^{(h)} = \Delta \in \mathbb{R}_+$*

**Assumption 4.** *We assume that the same state matrix $\mathbf{A}$ is used in S5 as in S4 (also cf. Assumption 2). Note this also specifies the S5 latent size $P = N$. We also assume the S5 input matrix is the horizontal concatenation of the column input vectors used by S4, $\mathbf{B} \triangleq \left[ \mathbf{B}^{(1)} \mid \ldots \mid \mathbf{B}^{(H)} \right]$.*

## D.2 DIFFERENT OUTPUT PROJECTIONS OF EQUIVALENT DYNAMICS

We provide a proof of Proposition 2.

**Proposition 2.** *Consider an S5 layer, with state matrix $\mathbf{A}$, input matrix $\mathbf{B}$ and some output matrix $\mathbf{C}$ (cf. Assumption 1); and an S4 layer, where each of the $H$ S4 SSMs has state matrix $\mathbf{A}$ (cf. Assumption 2, 4) and input vector $\mathbf{B}^{(h)}$ (cf. Assumption 4). If the S4 and S5 layers are discretized with the same timescales (cf. Assumption 3), then the S5 SSM produces outputs, $\mathbf{y}_k$, equivalent to a linear combination of the latent states of the $H$ S4 SSMs, $\mathbf{y}_k = \mathbf{C}^{\text{equiv}} \mathbf{x}_k^{(1:H)}$, where $\mathbf{C}^{\text{equiv}} = [\, \mathbf{C} \ \cdots \ \mathbf{C} \,]$.*

*Proof.* For a single S4 SSM, the discretized latent states can be expressed as a function of the input sequence $\mathbf{u}_{1:L} \in \mathbb{R}^{L \times H}$

$$\mathbf{x}_k^{(h)} = \sum_{i=1}^{k} \overline{\mathbf{A}}^{k-i} \overline{\mathbf{B}}^{(h)} u_i^{(h)}. \tag{13}$$

For an S5 layer, the latent states are expressible as

$$\mathbf{x}_k = \sum_{i=1}^{k} \overline{\mathbf{A}}^{k-i} \overline{\mathbf{B}} \mathbf{u}_i, \tag{14}$$

where we index as $\overline{\mathbf{B}} \triangleq \left[ \overline{\mathbf{B}}^{(1)} \mid \ldots \mid \overline{\mathbf{B}}^{(H)} \right]$ and $\mathbf{u}_i \triangleq \left[ u_i^{(1)}, \ldots, u_i^{(H)} \right]^{\top}$

Here we make the observation:

$$\mathbf{x}_k = \sum_{h=1}^{H} \mathbf{x}_k^{(h)}, \tag{15}$$

where this result follows directly from the linearity of (13) and (14). This shows that (under the assumptions outlined above) the states of the MIMO S5 SSM are equivalent to the summation of the states across the $H$ different SISO S4 SSMs.

We can then consider the effect of the output matrix $\mathbf{C}$. For S5, the output matrix is a single dense matrix

$$\mathbf{y}_k = \mathbf{C}\mathbf{x}_k. \tag{16}$$

We can substitute the relationship in (15) into (16) to cast the outputs of the MIMO S5 SSM in terms of the state of the $H$ SISO S4 SSMs:

$$\mathbf{y}_k = \mathbf{C}\sum_{h=1}^{H} \mathbf{x}_k^{(h)}, \tag{17}$$

$$= \sum_{h=1}^{H} \mathbf{C}\mathbf{x}_k^{(h)}. \tag{18}$$

Denoting the vertical concatenation of the $H$ S4 SSM state vectors $\mathbf{x}_k^{(1:H)} = \left[\mathbf{x}_k^{(1)^\top}, \ldots, \mathbf{x}_k^{(H)^\top}\right]^\top$, we see that the outputs of the S5 SSM are expressible as:

$$\mathbf{y}_k = \mathbf{C}^{\text{equiv}}\mathbf{x}_k^{(1:H)}, \quad \text{where} \quad \mathbf{C}^{\text{equiv}} = [\,\mathbf{C}\mid \cdots \mid \mathbf{C}\,], \tag{19}$$

and hence are equivalent to a linear combination of the $HN$ states computed by the $H$ S4 SSMs. $\quad\square$

This shows the outputs of the constrained S5 SSM under consideration (cf. Assumption 4) can be interpreted as a linear combination of the latent states computed by $H$ constrained S4 SSMs with the same state matrices and timescale parameters. Note however, it does *not* show that the outputs of the S5 SSM directly equal the outputs of the effective block-diagonal S4 SSM. Indeed, they are not equal, and we can repeat this analysis for the S4 layer to concretely identify the difference. For comparison we assume that the output vector for each S4 SSM is given as a row in the S5 output matrix, i.e. $\mathbf{C} = \left[\mathbf{C}^{(1)^\top}\mid \ldots \mid \mathbf{C}^{(H)^\top}\right]^\top$. We can express the output of each S4 SSM as

$$y_k^{(h)} = \mathbf{C}^{(h)}\mathbf{x}_k^{(h)}, \tag{20}$$

where $y_k^{(h)} \in \mathbb{R}$. We can then define the effective output matrix that operates on the entire latent space (the dashed box labelled $\mathbf{C}$ in Figure 2a) in S4 as

$$y_k^{(h)} = \left(\mathbf{C}^{\text{S4}}\mathbf{x}_k\right)^{(h)} \tag{21}$$

By inspecting (19) and (21), we can concretely express the difference in the equivalent output matrix used by both layers

$$\mathbf{C}^{\text{S4}} = \begin{bmatrix} \mathbf{C}^{(1)} & \cdots & \mathbf{0} \\ \vdots & \ddots & \vdots \\ \mathbf{0} & \cdots & \mathbf{C}^{(H)} \end{bmatrix}, \quad \mathbf{C}^{\text{equiv}} = \begin{bmatrix} \mathbf{C}^{(1)} & \cdots & \mathbf{C}^{(1)} \\ \vdots & \ddots & \vdots \\ \mathbf{C}^{(H)} & \cdots & \mathbf{C}^{(H)} \end{bmatrix} = [\,\mathbf{C}\mid \cdots \mid \mathbf{C}\,]. \tag{22}$$

In S4, the effective output matrix consists of independent vectors on the leading diagonal (as is pictured in Figure 2a). In contrast, the effective output matrix used by S5 instead ties dense output matrices across the $H$ S4 SSMs. As such, S5 can be interpreted as simply defining a different projection of the $H$ independent SISO SSMs than is used by S4. Note that both projection matrices have the same number of parameters.

Although the projection is different, the fact that the latent dynamics can still be *interpreted* as a linear projection of the same underlying S4 latent dynamics suggests that initializing the state dynamics in S5 with the HiPPO-LegS matrix may lead to good performance, similarly to what was observed in S4. We discuss this in the next section. We note that it is not obvious whether tying the dense output matrices is any more or less expressive than S4's use of a single untied output vector for each SSM, and it is unlikely that one approach is universally better than the other. We also stress that one would never implement S4 using the block diagonal matrix in (22), or, implement S5 using the repeated

matrix in (22). These matrices are simply constructs for understanding the equivalence between S4 and S5.

Finally, we note an alternative view: the block-diagonal S4 with output matrix $\mathbf{C}^{\text{equiv}}$ of Proposition 2 is equivalent to a version of S4 that uses a bank of single-input, multi-output (SIMO) SSMs with tied state matrices, timescales and multi-channel output matrices $\mathbf{C}^{(h)} = \mathbf{C} \in \mathbb{R}^{H \times N}$ and sums the individual SSM outputs. See appendix of Gu et al. (2021b) for a discussion of how the *linear state space layer* (LSSL), a predecessor of S4, can have multiple output channels.

### D.3    Diagonalizable Initialization

Proposition 2 suggests that initializing with the HiPPO-LegS matrix may yield good performance in S5, just as it does in S4 (because the constrained version of S5 under consideration is effectively a different linear projection of the same latent dynamics). However, the HiPPO-LegS matrix is not stably diagonalizable. Corollary 1 allows us to initialize MIMO SSMs with the diagonalizable HiPPO-N matrix to approximate the HiPPO-LegS matrix and expect the performance to be comparable.

**Corollary 1** (Extension of Theorem 3 in Gu et al. (2022)). *Consider $\mathbf{A}_{\text{LegS}} \in \mathbb{R}^{N \times N}$, $\mathbf{A}_{\text{LegS}}^{\text{Normal}} \in \mathbb{R}^{N \times N}$, $\mathbf{B}_{\text{LegS}} \in \mathbb{R}^{N \times H}, \mathbf{P}_{\text{LegS}} \in \mathbb{R}^{N}$ as defined in Appendix B.1.1. Given vector-valued inputs $\mathbf{u}(t) \in \mathbb{R}^{H}$, the ordinary differential equation $\frac{\mathrm{d}\mathbf{x}'(t)}{\mathrm{d}t} = \mathbf{A}_{\text{LegS}}^{\text{Normal}}\mathbf{x}'(t) + \frac{1}{2}\mathbf{B}_{\text{LegS}}\mathbf{u}(t)$ converges to $\frac{\mathrm{d}\mathbf{x}(t)}{\mathrm{d}t} = \mathbf{A}_{\text{LegS}}\mathbf{x}(t) + \mathbf{B}_{\text{LegS}}\mathbf{u}(t)$ as $N \to \infty$.*

*Proof.* Theorem 3 in Gu et al. (2022) shows the following relationship for scalar input signals as $N \to \infty$:

$$\frac{\mathrm{d}\mathbf{x}^{(h)}(t)}{\mathrm{d}t} = \mathbf{A}_{\text{LegS}}^{\text{Normal}}\mathbf{x}^{(h)}(t) + \frac{1}{2}\mathbf{B}_{\text{LegS}}^{(h)}u^{(h)}(t), \tag{23}$$

where the only modification we have made is introducing the $(h)$ superscript to allow us to explicitly index dimensions later. We define $\mathbf{B} = \left[\mathbf{B}_{\text{LegS}}^{(1)} \mid \dots \mid \mathbf{B}_{\text{LegS}}^{(H)}\right]$. We wish to extend this to the case of vector-valued input signals.

We first recall (15), which shows that the latent states of the MIMO S5 SSM are the summation of the latent states of the $H$ SISO S4 SSMs (to which Theorem 3 from Gu et al. (2021a) applies). Although we derived (15) in discrete time, it applies equally in continuous time:

$$\mathbf{x}(t) = \sum\nolimits_{h=1}^{H} \mathbf{x}^{(h)}(t). \tag{24}$$

We can therefore define the derivative of the S5 state as:

$$\frac{\mathrm{d}\mathbf{x}(t)}{\mathrm{d}t} = \sum\nolimits_{h=1}^{H} \frac{\mathrm{d}\mathbf{x}^{(h)}(t)}{\mathrm{d}t}. \tag{25}$$

Substituting (23) into this then yields:

$$\frac{\mathrm{d}\mathbf{x}(t)}{\mathrm{d}t} = \sum\nolimits_{h=1}^{H} \left[\mathbf{A}_{\text{LegS}}^{\text{Normal}}\mathbf{x}^{(h)}(t) + \frac{1}{2}\mathbf{B}_{\text{LegS}}^{(h)}u^{(h)}(t)\right], \tag{26}$$

$$= \mathbf{A}_{\text{LegS}}^{\text{Normal}} \sum\nolimits_{h=1}^{H} \mathbf{x}^{(h)}(t) + \frac{1}{2} \sum\nolimits_{h=1}^{H} \mathbf{B}_{\text{LegS}}^{(h)}u^{(h)}(t), \tag{27}$$

$$= \mathbf{A}_{\text{LegS}}^{\text{Normal}}\mathbf{x}(t) + \frac{1}{2}\mathbf{B}_{\text{LegS}}\mathbf{u}(t). \tag{28}$$

$\square$

This equivalence motivates initializing S5 state matrices with the diagonalizable HiPPO-N matrix and suggests that we can expect to see similar performance gains.

### D.4 RELAXING THE ASSUMPTIONS

Here we discuss how relaxing the constraint on S5's latent size from Assumption 4 helps to relax the assumptions on the tied S4 SSM state matrices (Assumption 2) and timescales (Assumption 3) as well as the tied output matrices that result from Proposition 2.

We start by considering the case when the S5 SSM state matrix is block-diagonal. Consider an S5 SSM with latent size $JN = \mathcal{O}(H)$ and block-diagonal $\mathbf{A} \in \mathbb{R}^{JN \times JN}$, dense $\mathbf{B} \in \mathbb{R}^{JN \times H}$, dense $\mathbf{C} \in \mathbb{R}^{H \times JN}$, and $J$ different timescale parameters $\mathbf{\Delta} \in \mathbb{R}^{J}$. As a result of the block-diagonal state matrix, this system has a latent state $\mathbf{x}_k \in \mathbb{R}^{JN}$ that can be partitioned into $J$ different states $\mathbf{x}_k^{(j)} \in \mathbb{R}^{N}$. We can then partition this system into $J$ different subsystems and discretize each subsystem with one of the $\Delta^{(j)}$ to get the following discretized system:

$$\overline{\mathbf{A}} = \begin{bmatrix} \overline{\mathbf{A}}^{(1)} & & \\ & \ddots & \\ & & \overline{\mathbf{A}}^{(J)} \end{bmatrix}, \quad \overline{\mathbf{B}} = \begin{bmatrix} \overline{\mathbf{B}}^{(1)} \\ \vdots \\ \overline{\mathbf{B}}^{(J)} \end{bmatrix}, \quad \mathbf{C} = \begin{bmatrix} \mathbf{C}^{(1)} \mid \cdots \mid \mathbf{C}^{(J)} \end{bmatrix}, \quad (29)$$

where $\overline{\mathbf{A}}^{(j)} \in \mathbb{R}^{N \times N}, \overline{\mathbf{B}}^{(j)} \in \mathbb{R}^{N \times H}$ and $\mathbf{C}^{(j)} \in \mathbb{R}^{H \times N}$. It follows that this partitioned system can be also be viewed as $J$ independent $N$-dimensional S5 SSM subsystems and the output of the overall system is the sum of the output of the $J$ subsystems

$$\mathbf{y}_k = \mathbf{C}\mathbf{x}_k \quad (30)$$

$$= \sum_{j=1}^{J} \mathbf{C}^{(j)} \mathbf{x}_k^{(j)}. \quad (31)$$

It follows from Proposition 2 that the dynamics of each of these $J$ S5 SSM subsystems can be related to the dynamics of a *different* S4 system from Proposition 2. Each of these S4 systems has its own bank of tied S4 SSMs (cf. Assumptions 2, 3). Importantly, each of the $J$ S4 systems can have its own state matrix, timescale parameter and output matrix shared across its $H$ S4 SSMs. Thus, the outputs of a $JN$ dimensional S5 SSM can be equivalent to the linear combination of the latent states of $J$ different S4 systems from Proposition 2. This fact motivates the option to initialize a block-diagonal S5 state matrix with several HiPPO-N matrices on the blocks, rather than just initializing with one larger HiPPO-N matrix. In practice we found the block-diagonal initialization to improve performance on many tasks, see Appendix E.

### D.5 TIMESCALE PARAMETERIZATION

Finally, we take a closer look at the parameterization of the timescale parameters $\mathbf{\Delta}$. As discussed in Section 4.3, S4 can learn a different timescale parameter for each S4 SSM, potentially allowing it to capture different timescales of the data. Further, the initialization of the timescales can be important (Gu et al., 2023; Gupta et al., 2022), and limiting to sampling a single initial parameter may lead to poor initialization. The discussion in the previous section motivates potentially learning $J$ different timescale parameters, one for each of the $J$ subsystems. However, in practice, we found better performance when using $P$ different timescale parameters, one for each of the states. On the one hand, this can be viewed simply as learning a different scaling for each of the eigenvalues in the diagonalized system (see Eq. (6)). On the other hand, this could be viewed as increasing the number of timescale parameters sampled at initialization, helping to combat the possibility of poor initialization. Of course, the system could *learn* to use just a single timescale by setting all of the timescales to be the same. See further discussion in the ablation study in Appendix E.

# E   ABLATIONS

We perform several ablations to empirically explore different aspects of S5.

## E.1   S5 LATENT SIZE, BLOCK-DIAGONAL INITIALIZATION, AND TIMESCALE PARAMETERIZATION

The discussion in Section 4 and Appendix D raises several interesting questions: How does S5 perform when the latent size $P$ is restricted to be equal to the latent size $N$ used by each of S4's SSMs? How important is the timescale parameterization discussed in Appendix D.5? How important is the block-diagonal initialization? Table 5 displays the results of an ablation study performed on the LRA tasks to get a better sense of this. We consider 3 versions of S5.

The first version uses the same general architecture (e.g. number of input/output features $H$, number of layers, etc) as reported for the S4/S4D variants in Gu et al. (2022), sets the S5 SSM latent size $P$ to be equal to the latent size $N = 64$ used by each of the S4 SSMs, and uses only a single scalar timescale parameter $\Delta \in \mathbb{R}$. This is essentially the version of S5 we consider in Proposition 2. We observe that this constrained version of S5 actually performs well on most tasks, though struggles to perform comparably to the S4 baselines on Image and ListOps.

The second version of S5 is exactly the same as the first except we parameterize the timescale parameter as a vector $\Delta \in \mathbb{R}^N$. We observe uniform improvements over the scalar timescale parameterization and this reflects our general findings when training S5.

Finally, the complexity analysis and runtime comparison in Appendix C.2 suggests the latent size of S5 can be increased while maintaining similar complexity and practical runtimes as S4. We include the unconstrained version of S5 reported in our main results that uses the settings reported in the hyperparameter Table 11. These models were allowed to be parameterized with fewer input/output features $H$ (to ensure similar parameter counts to the S4 baselines) and generally used larger latent sizes $P > N$. Further, we swept over the use of a block-diagonal initialization or not and the number of blocks to use (where $J = 1$ indicates no block-diagonal initialization was used). All models benefited from the block-diagonal initialization for the LRA tasks (See Table 11 ).

Table 5: Ablations on the LRA benchmark tasks (Tay et al., 2021). S4 results were taken from Gu et al. (2022; 2021a). Note that the total parameter count for all models in this table are commensurate, and hence variations in performance cannot be attributed to a model having drastically more parameters.

| Model (Input length) | ListOps (2,048) | Text (4,096) | Retrieval (4,000) | Image (1,024) | Pathfinder (1,024) | Path-X (16,384) |
|---|---|---|---|---|---|---|
| S4D-LegS | 60.47 | 86.18 | 89.46 | 88.19 | 93.06 | 91.95 |
| S4-LegS | 59.60 | 86.82 | 90.90 | **88.65** | 94.20 | 96.35 |
| S5 (P=N, $J = 1$, $\Delta \in \mathbb{R}$) | 57.20 | 87.60 | 90.53 | 82.01 | 94.14 | 96.59 |
| S5 (P=N, $J = 1$, $\boldsymbol{\Delta} \in \mathbb{R}^N$) | 58.65 | 88.12 | 90.76 | 85.04 | 94.53 | 97.49 |
| S5 (P: free, J≥1, $\boldsymbol{\Delta} \in \mathbb{R}^P$, see Table 11) | **62.15** | **89.31** | **91.40** | 88.00 | **95.33** | **98.58** |

## E.2   IMPORTANCE OF HiPPO-N AND CONTINUOUS-TIME PARAMETERIZATION

We perform a further ablation study to gain insight into the differences between S5 and prior attempts at parallelized linear RNNs (discussed in Section 5) focusing on what appears to be the distinguishing features: continuous-time parameterizations and HiPPO initializations. We compare different initializations of the state matrix: random Gaussian, random antisymmetric, and HiPPO-N. The antisymmetric initialization is interesting because prior work considered these matrices in RNNs for long-range dependencies (Chang et al., 2019), and because the HiPPO-LegS matrix can be parameterized in a way related to antisymmetric matrices (Gu et al., 2021a). Moreover, to compare to a setup more akin to the previous parallelized linear RNN work, we also consider a direct discrete-time parameterization of S5 that does not perform repeated discretization during training or learn the

Table 6: S5 Initialization and Parameterization Ablation Study. ✗ indicates the model did not improve over random guessing.

| Model (Input length) | Parameterization | Initialization | ListOps (2,048) | Text (4,096) | Path-X (16,384) |
|---|---|---|---|---|---|
| S5 (ablation) | Discrete | Gaussian | 41.50 | 81.09 | ✗ |
| S5 (ablation) | Discrete | Antisymmetric | 49.10 | 86.42 | ✗ |
| S5 (ablation) | Discrete | HiPPO-N | 58.15 | 61.93 | ✗ |
| S5 (ablation) | Continuous | Gaussian | 58.50 | 69.03 | ✗ |
| S5 (ablation) | Continuous | Antisymmetric | 59.35 | 82.83 | ✗ |
| **S5** | **Continuous** | **HiPPO-N** | **62.15** | **89.31** | **98.58** |

timescale parameter $\Delta$. We present the results of this ablation study in Table 6 (along with S5). We consider three of the LRA tasks that vary in length and difficulty.

The main takeaway is that the only approach that consistently performs well on all tasks, including the ability to solve Path-X, is the S5 approach that uses the continuous-time parameterization and HiPPO initialization. We also note that we observed the discrete time/HiPPO-N matrix configuration to be difficult to train due to stability issues, typically requiring a much lower learning rate.

### E.3 S4D INITIALIZATION ABLATIONS

Finally, Gu et al. (2022) propose several alternative diagonal matrices to the diagonalized HiPPO-N matrix, including the S4D-Inv and S4D-Lin matrices. They perform an ablation on the LRA tasks where they simply replace the diagonalized HiPPO-N matrix with the S4D-Inv and S4D-Lin matrices while keeping all other factors the same. We include these results in Table 7. In Table 7, we also include results for a similar ablation in S5 by using these matrices to initialize S5 in place of the HiPPO-N matrix while keeping all other factors constant. Both matrices perform well on most tasks with the exception of the S4D-Lin matrix on Path-X. Interestingly, one of these runs reached 96.79%, however the other runs did not exceed random guessing on this task. Future exploration of these and other matrices are an interesting direction for future work.

## F Supplementary Results

We include further experimental results to supplement the results presented in the main text.

### F.1 Extended LRA Results

Table 7: Test accuracy on the LRA benchmark tasks (Tay et al., 2021). ✗ indicates the model did not exceed random guessing. Citations refer to the original model. The results for Transformer through Performer are from (Tay et al., 2021). We follow the procedure reported in Gu et al. (2021a; 2022) and report means across three seeds for S4, S4D (as reported by Gu et al. (2021a; 2022)) and S5, with the standard deviation in parenthesis.

| Model (Input length) | ListOps (2,048) | Text (4,096) | Retrieval (4,000) | Image (1,024) | Pathfinder (1,024) | Path-X (16,384) | Avg. |
|---|---|---|---|---|---|---|---|
| Transformer (Vaswani et al., 2017) | 36.37 | 64.27 | 57.46 | 42.44 | 71.40 | ✗ | 53.66 |
| Reformer (Kitaev et al., 2020) | 37.27 | 56.10 | 53.40 | 38.07 | 68.50 | ✗ | 50.56 |
| BigBird (Zaheer et al., 2020) | 36.05 | 64.02 | 59.29 | 40.83 | 74.87 | ✗ | 54.17 |
| Linear Trans. (Katharopoulos et al., 2020) | 16.13 | 65.90 | 53.09 | 42.34 | 75.30 | ✗ | 50.46 |
| Performer (Choromanski et al., 2021) | 18.01 | 65.40 | 53.82 | 42.77 | 77.05 | ✗ | 51.18 |
| FNet (Lee-Thorp et al., 2022) | 35.33 | 65.11 | 59.61 | 38.67 | 77.80 | ✗ | 54.42 |
| Nyströmformer (Xiong et al., 2021) | 37.15 | 65.52 | 79.56 | 41.58 | 70.94 | ✗ | 57.46 |
| Luna-256 (Ma et al., 2021) | 37.25 | 64.57 | 79.29 | 47.38 | 77.72 | ✗ | 59.37 |
| H-Transformer-1D (Zhu & Soricut, 2021) | 49.53 | 78.69 | 63.99 | 46.05 | 68.78 | ✗ | 61.41 |
| CCNN (Romero et al., 2022a) | 43.60 | 84.08 | ✗ | 88.90 | 91.51 | ✗ | 68.02 |
| Mega ($\mathcal{O}(L^2)$) (Ma et al., 2023) | **63.14** | **90.43** | 91.25 | **90.44** | **96.01** | 97.98 | **88.21** |
| Mega-chunk ($\mathcal{O}(L)$) (Ma et al., 2023) | 58.76 | 90.19 | 90.97 | 85.80 | 94.41 | 93.81 | 85.66 |
| DSS$_{\text{SOFTMAX}}$ (Gupta et al., 2022) | 60.60 | 84.80 | 87.80 | 85.70 | 84.60 | 87.80 | 81.88 |
| S4D-LegS (Gu et al., 2022) | 60.47 (0.34) | 86.18 (0.43) | 89.46 (0.14) | 88.19 (0.26) | 93.06 (1.24) | 91.95 | 84.89 |
| S4D-Inv (Gu et al., 2022) | 60.18 (0.35) | 87.34 (0.20) | 91.09 (0.01) | 87.83 (0.37) | 93.78 (0.25) | 92.80 | 85.50 |
| S4D-Lin (Gu et al., 2022) | 60.52 (0.51) | 86.97 (0.23) | 90.96 (0.09) | 87.93 (0.34) | 93.96 (0.60) | ✗ | 78.39 |
| S4-FouT (Gu et al., 2023) | 57.88 (1.90) | 86.34 (0.31) | 89.66 (0.88) | 89.07 (0.19) | 94.46 (0.24) | ✗ | 77.90 |
| S4-LegS/FouT (Gu et al., 2023) | 60.45 (0.75) | 86.78 (0.26) | 90.30 (0.28) | 89.00 (0.26) | 94.44 (0.08) | ✗ | 78.50 |
| S4-LegS (Gu et al., 2021a) | 59.60 (0.07) | 86.82 (0.13) | 90.90 (0.15) | 88.65 (0.23) | 94.20 (0.25) | 96.35 | 86.09 |
| Liquid-S4 (Hasani et al., 2023) | 62.75 (0.20) | 89.02 (0.04) | 91.20 (0.01) | 89.50 (0.40) | 94.80 (0.2) | 96.66 (0.001) | 87.32 |
| S5-Inv (See Appendix E.3) | 60.07 (0.26) | 87.77 (0.29) | 91.26 (0.12) | 86.41 (0.17) | 93.42 (0.42) | 97.54 (0.74) | 86.08 |
| S5-Lin (See Appendix E.3) | 59.98 (0.53) | 88.15 (0.24) | 91.31 (0.24) | 86.05 (0.96) | 94.31 (0.36) | 65.60 (27.00) | 80.90 |
| **S5** | 62.15 (0.23) | 89.31 (0.15) | **91.40** (0.05) | 88.00 (0.22) | 95.33 (0.26) | **98.58** (0.17) | 87.46 |

## F.2 EXTENDED SPEECH RESULTS

Table 8: Speech Commands classification task (Warden, 2018). Test accuracy on 35-way keyword spotting. Training examples are one-second audio waveforms sampled at 16kHz, or a one-dimensional sequence of length 16000. Last column indicates zero-shot testing at 8kHz where examples are constructed by naive decimation. The mean across three random seeds is reported, with the standard deviation in parenthesis. Performance for the baselines InceptionNet through to S4D-Lin are reported from Gu et al. (2022).

| Model | Parameters | 16000Hz | 8000Hz |
|---|---|---|---|
| InceptionNet (Nonaka & Seita, 2021) | 481K | 61.24 (0.69) | 05.18 (0.07) |
| ResNet-18 (Nonaka & Seita, 2021) | 216K | 77.86 (0.24) | 08.74 (0.57) |
| XResNet-50 (Nonaka & Seita, 2021) | 904K | 83.01 (0.48) | 07.72 (0.39) |
| ConvNet (Nonaka & Seita, 2021) | 26.2M | 95.51 (0.18) | 07.26 (0.79) |
| S4-LegS (Gu et al., 2021a) | 307K | 96.08 (0.15) | 91.32 (0.17) |
| S4-FouT (Gu et al., 2023) | 307K | 95.27 (0.20) | 91.59 (0.23) |
| S4-(LegS/FouT) (Gu et al., 2023) | 307K | 95.32 (0.10) | 90.72 (0.68) |
| S4D-LegS (Gu et al., 2022) | 306K | 95.83 (0.14) | 91.08 (0.16) |
| S4D-Inv (Gu et al., 2022) | 306K | 96.18 (0.27) | 91.80 (0.24) |
| S4D-Lin (Gu et al., 2022) | 306K | 96.25 (0.03) | 91.58 (0.33) |
| Liquid-S4 (Hasani et al., 2023) | 224K | **96.78** (0.05) | 90.00 (0.25) |
| **S5** | 280K | 96.52 (0.16) | **94.53** (0.10) |

### F.3 Pendulum Extended Results

We also evaluate two ablations: *S5-drop* uses the same S5 architecture, but drops the dependence on the inter-sample interval, i.e. $\Delta_t \triangleq 1.0$. We expect this network to perform poorly as it has no knowledge of how long has elapsed between observations. *S5-append* uses the same S5 architecture, but appends the integration timestep to the thirty-dimensional image encoding, prior to being input into the dense S5 input layer. Hypothetically, we expect this network to perform as well as S5. However, to do so, requires the S5 network to learn to process time, which may be difficult, especially in more complex domains. We include these ablations in the bottom partition of Table 9.

Note that the runtimes quoted for the baseline methods (runtimes marked with a *) are as reported by Schirmer et al. (2022). These times are the total time for a training epoch, and hence include any time spent batching data. We re-ran the CRU using the original PyTorch code on the same hardware as we run our JAX S5 experiments on (labelled *CRU (our run)*). For these experiments we used a single NVIDIA GeForce RTX 2080 Ti. For these runs (*CRU (our run)*, *S5*, *S5-drop* and *S5-append*) we exclude the time spent batching the data to more faithfully compare the runtimes for the models themselves. Also note that our S5 experiments will benefit from JAX compilation, but that this is not sufficient to explain the difference in runtime.

Table 9: Test MSE $\times 10^{-3}$ and runtimes on the pendulum regression task. Performance for the baselines, mTAND through to CRU, are reported from Schirmer et al. (2022), with mean and standard deviations across five random seeds (standard deviation in parenthesis). Accompanying citation indicates the original citation for the method. We re-ran the CRU (labelled *CRU (our run)*) and ran our S5 methods across twenty random seeds. We report mean and variances of the MSE error on the held-out test set, using a model selected using the validation set MSE. We refer the reader to Schirmer et al. (2022) for full description of the baselines.

| Model | Runtime (sec/train epoch) | Regression MSE ($\times 10^{-3}$) |
|---|---|---|
| mTAND (Shukla & Marlin, 2021) | 3* | 65.64 (4.05) |
| RKN (Becker et al., 2019) | 20* | 8.43 (0.61) |
| RKN-$\Delta_t$ (Becker et al., 2019) | 20* | 5.09 (0.40) |
| GRU (Cho et al., 2014) | 12* | 9.44 (1.00) |
| GRU-$\Delta_t$ (Cho et al., 2014) | 12* | 5.44 (0.99) |
| Latent ODE (Chen et al., 2018) | 52* | 15.70 (2.85) |
| ODE-RNN (Rubanova et al., 2019) | 37* | 7.26 (0.41) |
| GRU-ODE-B (De Brouwer et al., 2019) | 60* | 9.78 (3.40) |
| f-CRU (Schirmer et al., 2022) | 29* | 6.16 (0.88) |
| CRU (Schirmer et al., 2022) | 36* | 4.63 (1.07) |
| CRU (our run) | 22 | 3.94 (0.21) |
| **S5** | 0.25 | **3.41** (0.27) |
| S5-drop (ablation) | **0.25** | 6.68 (0.38) |
| S5-append (ablation) | 0.25 | 4.13 (0.43) |

## F.4 PIXEL-LEVEL 1-D IMAGE CLASSIFICATION RESULTS

Table 10 presents results and citations of the pixel-level 1-D image classification.

Table 10: Test accuracy on Pixel-level 1-D Image classification. Citations refer to the original model; additional citation indicates work from which this baseline is reported.

| Model (Input length) | sMNIST (784) | psMNIST (784) | sCIFAR (1024) |
|---|---|---|---|
| Transformer (Trinh et al., 2018; Vaswani et al., 2017) | 98.9 | 97.9 | 62.2 |
| CCNN (Romero et al., 2022a) | **99.72** | **98.84** | **93.08** |
| FlexTCN (Romero et al., 2021) | 99.62 | 98.63 | 80.82 |
| CKConv (Romero et al., 2022b) | 99.32 | 98.54 | 63.74 |
| TrellisNet (Bai et al., 2019) | 99.20 | 98.13 | 73.42 |
| TCN (Bai et al., 2018) | 99.0 | 97.2 | - |
| LSTM (Gu et al., 2020b; Hochreiter & Schmidhuber, 1997) | 98.9 | 95.11 | 63.01 |
| r-LSTM (Trinh et al., 2018) | 98.4 | 95.2 | 72.2 |
| Dilated GRU (Chang et al., 2017) | 99.0 | 94.6 | - |
| Dilated RNN (Chang et al., 2017) | 98.0 | 96.1 | - |
| IndRNN (Li et al., 2018) | 99.0 | 96.0 | - |
| expRNN (Lezcano-Casado & Martınez-Rubio, 2019) | 98.7 | 96.6 | - |
| UR-LSTM (Gu et al., 2020b) | 99.28 | 96.96 | 71.00 |
| UR-GRU (Gu et al., 2020b) | 99.27 | 96.51 | 74.4 |
| LMU (Voelker et al., 2019) | - | 97.15 | - |
| HiPPO-RNN (Gu et al., 2020a) | 98.9 | 98.3 | 61.1 |
| UNIcoRNN (Rusch & Mishra, 2021) | - | 98.4 | - |
| LMU-FFT (Chilkuri & Eliasmith, 2021) | - | 98.49 | - |
| LipschitzRNN (Erichson et al., 2021) | 99.4 | 96.3 | 64.2 |
| LSSL (Gu et al., 2021b) | 99.53 | 98.76 | 84.65 |
| S4 (Gu et al., 2022; 2021a) | 99.63 | 98.70 | 91.80 |
| S4D (Gu et al., 2022) | - | - | 89.92 |
| Liquid-S4 (Hasani et al., 2023). | - | - | 92.02 |
| **S5** | 99.65 | 98.67 | 90.10 |

## G  EXPERIMENT CONFIGURATIONS

In this section we describe the experimental details. This includes the model architecture, general hyperparameters, specifics for each task, and information about the datasets.

### G.1  DEEP SEQUENCE MODEL ARCHITECTURE

For the experiments, we use the S5 layer as a drop-in replacement for the S4 layer used in the sequence model architecture of (Gu et al., 2021a). On a high level, this architecture consists of a linear encoder (to encode the input at each time step into $H$ features), multiple S5 layers, a mean pooling layer, a linear decoder, and a Softmax operation for the classification tasks. The mean pooling layer compresses the output of the last S5 layer, of shape [batch size, sequence length, number of features ($H$)], across the sequence length dimension, so that a single $H$-dimensional encoding is available for softmax classification.

The baseline S4 models from Gu et al. (2022; 2023) apply a GLU activation (Dauphin et al., 2017) function to the S4 SSM outputs. To take advantage of the fact that the S5 SSM outputs have already been mixed throughout the MIMO SSM we use what is essentially a weighted sigmoid gated unit (Tanaka, 2020) (a GLU activation without an additional linear transform). Specifically, given an S5 SSM output $\mathbf{y}_k \in \mathbb{R}^H$ and a dense matrix $\mathbf{W} \in \mathbb{R}^{H \times H}$, the layer output of the activation function we apply is $\mathbf{u}'_k = \text{GELU}(\mathbf{y}_k) \odot \sigma(\mathbf{W} * \text{GELU}(\mathbf{y}_k))$.

Hyperparameter options such as dropout rate, using either layer normalization or batch normalization, and using either pre-norm or post-norm are applied between the layers. Exceptions to the basic architecture described here are mentioned in the individual experiment sections below.

### G.2  DEFAULT HYPERPARAMETERS

Table 11 presents the main hyperparameters used for each experiment. For all experiments we ensure the number of layers and layer input/output features $H$ are less than or equal to the number of layers and layer input/output features reported in Gu et al. (2022) as well as ensuring comparable parameter counts.

In general, the models for most tasks used batch normalization and pre-norm. Exceptions are noted in the individual experiment sections below.

#### G.2.1  OPTIMIZERS AND LEARNING RATES

We follow the general optimization approach used by S4/S4D in Gu et al. (2022). We use the AdamW optimizer (Loshchilov & Hutter, 2019) with a global learning rate. However, in general, no weight decay and a smaller learning rate (the SSM learning rate) is applied to $\mathbf{\Lambda}, \tilde{\mathbf{B}}, \mathbf{\Delta}$. All experiments used cosine annealing. Exceptions to these points are noted in the individual experiment sections below.

#### G.2.2  BIDIRECTIONALITY

We follow Gu et al. (2022) and use bidirectional models for the LRA and speech tasks. Unidirectional (causal) models were used for the pendulum, sequential and permuted MNIST for fair comparison with prior methods that used unidirectional models.

### G.3  TASK SPECIFIC HYPERPARAMETERS

Here we specify any task-specific details, hyperparameter or architectural differences from the defaults outlined above.

#### G.3.1  LISTOPS

Weight decay and the global learning rate were applied to $\tilde{\mathbf{B}}$.

Table 11: Hyperparameters used for the reported results. Depth: number of layers. H: number of input/output features. P: Latent size. J: number of blocks used for the initialization of $\mathbf{A}$ (see Section B.1.1). Dropout: dropout rate. LR: global learning rate. SSM LR: the SSM learning rate. B: batch size. Epochs: max epochs set for the run. WD: weight decay.

|  | Depth | H | P | J | Dropout | LR | SSM LR | B | Epochs | WD |
|---|---|---|---|---|---|---|---|---|---|---|
| ListOps | 8 | 128 | 16 | 8 | 0.0 | 0.003 | 0.001 | 50 | 40 | 0.04 |
| Text | 6 | 256 | 192 | 12 | 0.1 | 0.004 | 0.001 | 50 | 35 | 0.07 |
| Retrieval | 6 | 128 | 256 | 16 | 0.0 | 0.002 | 0.001 | 32 | 20 | 0.05 |
| Image | 6 | 512 | 384 | 3 | 0.1 | 0.005 | 0.001 | 50 | 250 | 0.07 |
| Pathfinder | 6 | 192 | 256 | 8 | 0.05 | 0.005 | 0.0009 | 64 | 200 | 0.03 |
| Path-X | 6 | 128 | 256 | 16 | 0.0 | 0.002 | 0.0006 | 32 | 75 | 0.06 |
| Speech | 6 | 96 | 128 | 16 | 0.1 | 0.008 | 0.002 | 16 | 40 | 0.04 |
| Pendulum | 4 | 30 | 16 | 8 | 0.0 | 0.012 | 0.003 | 32 | 100 | 0.0 |
| sMNIST | 4 | 96 | 128 | 1 | 0.1 | 0.008 | 0.002 | 50 | 150 | 0.01 |
| psMNIST | 4 | 128 | 128 | 2 | 0.15 | 0.004 | 0.001 | 50 | 150 | 0.01 |
| sCIFAR | 6 | 512 | 384 | 3 | 0.1 | 0.0045 | 0.001 | 50 | 250 | 0.07 |

### G.3.2    TEXT

No exceptions to the defaults for this run.

### G.3.3    RETRIEVAL

This document matching task requires a slightly different architecture from the other experiments, as discussed in Tay et al. (2021). We use the same configuration as S4 (Gu et al., 2021a). Each string is passed through the input encoder, S5 layers, and mean pooling layers. Denoting $X_1$ as the output for the first document and $X_2$ as the output for the second document, four features are created and concatenated together (Tay et al., 2021) as

$$X = [X_1, X_2, X_1 * X_2, X_1 - X_2]. \tag{32}$$

This concatenated feature is then fed to a linear decoder and softmax function as normal.

### G.3.4    IMAGE

Weight decay and the global learning rate were applied to $\tilde{\mathbf{B}}$.

### G.3.5    PATHFINDER

No exceptions to the defaults for this run.

### G.3.6    PATH-X

Weight decay was applied to $\tilde{\mathbf{B}}$.

### G.3.7    SPEECH COMMANDS

No weight decay and the SSM learning rate were applied to $\tilde{\mathbf{C}}$ for this run.

### G.3.8    PENDULUM REGRESSION

We use the same encoder-decoder architecture as Schirmer et al. (2022). The encoder has layers: convolution, ReLU, max pool, convolution, ReLU, max pool, dense, ReLU, dense. The first convolution layer has twelve features, a $5 \times 5$ kernel, and a padding of $(2, 2)$. The second convolution layer has twelve features, a $3 \times 3$ kernel, a stride of 2, and a padding of $(1, 1)$. Both max pools use a window size of $2 \times 2$ and a stride of 2. The dense layer has thirty hidden units. The linear readout layer outputs $H = 30$ features. This is chosen to match the encoding size in Schirmer et al. (2022),

and is used for all layers. Separate mean and unconstrained variance decoders are used, defined as a one-layer MLP with a hidden size of thirty. An elu+1 activation function is used to constrain the variance to be positive.

Layer normalization and post-norm were used for this task.

For the timings presented in Table 3 and 9 we use a batch size of 50, instead of the batch size of 32 used during training, to match the batch sizes reported by the baselines.

### G.3.9 SEQUENTIAL MNIST

No exceptions to the defaults for this run.

### G.3.10 PERMUTED SEQUENTIAL MNIST

Weight decay and the global learning rate were applied to $\tilde{\mathbf{B}}$. Post-norm was used for this task.

### G.3.11 SEQUENTIAL CIFAR

We trained a model with the exact hyperparameter settings as used for the LRA-IMAGE (grayscale sequential CIFAR) task with no further tuning.

### G.4 DATASET DETAILS

We provide more context and details for each of the LRA (Tay et al., 2021) and Speech Commands (Warden, 2018) datasets we consider. Note that we follow the same data pre-processing steps as Gu et al. (2021a), which we also include here for completeness.

- `ListOps`: A lengthened version of the dataset presented by Nangia & Bowman (2018). Given a nested set of mathematical operations (such as `min` and `max`) and integer operands in the range zero to nine, expressed in prefix notation with brackets, compute the integer result of the mathematical expression, e.g. $[\mathrm{MAX}29[\mathrm{MIN}47]0] \rightarrow 9$. Characters are encoded as one-hot vectors, with 17 unique values possible (opening brackets and operators are grouped into a single token). The sequences are of unequal length, and hence the end of shorter sequences is padded with a fixed indicator value, padded to a maximum length of $2,000$. A reserved end-of-sequence token is appended. There are 10 different classes, representing the integer result of the expression. There are $96,000$ training sequences, $2,000$ validation sequences, and $2,000$ test sequences. No normalization is applied.

- `Text`: Based off of the iMDB sentiment dataset presented by Maas et al. (2011). Given a movie review, where characters are encoded as a sequence of integer tokens, classify whether the movie review is positive or negative. Characters are encoded as one-hot vectors, with 129 unique values possible. Sequences are of unequal length, and are padded to a maximum length of $4,096$. There are two different classes, representing positive and negative sentiment. There are $25,000$ training examples and $25,000$ test examples. No validation set is provided. No normalization is applied.

- `Retrieval`: Based off of the ACL Anthology network corpus presented by Radev et al. (2009). Given two textual citations, where characters are encoded as a sequence of integer tokens, classify whether the two citations are equivalent. The citations must be compressed separately, before being passed into a final classifier layer. This is to evaluate how effectively the network can represent the text. The decoder head then uses the encoded representation to complete the task. Characters are encoded into a one-hot vector with 97 unique values. Two paired sequences may be of unequal length, with a maximum sequence length of $4,000$. There are two different classes, representing whether the citations are equivalent or not. There are $147,086$ training pairs, $18,090$ validation pairs, and $17,437$ test pairs. No normalization is applied.

- `Image`: Uses the CIFAR-10 dataset presented by Krizhevsky (2009). Given a $32 \times 32$ grayscale CIFAR-10 image as a one-dimensional raster scan, classify the image into one of ten classes. Sequences are of equal length ($1,024$). There are ten different classes. There are $45,000$ training examples, $5,000$ validation examples, and $10,000$ test examples. RGB

pixel values are converted to a grayscale intensities, which are then normalized to have zero mean and unit variance (across the entire dataset).

- `Pathfinder`: Based off of the Pathfinder challenge introduced by Linsley et al. (2018). A $32 \times 32$ grayscale image image shows a start and an end point as a small circle. There are a number of dashed lines on the image. The task is to classify whether there is a dashed line (or path) joining the start and end point. There are two different classes, indicating whether there is a valid path or not. Sequences are all of the same length ($1,024$). There are $160,000$ training examples, $20,000$ validation examples, and $20,000$ test examples. The data is normalized to be in the range $[-1, 1]$.

- `Path-X`: An "extreme" version of the `Pathfinder` challenge. Instead, the images are $128 \times 128$ pixels, resulting in sequences that are a factor of sixteen times longer. Otherwise identical to the `Pathfinder` challenge.

- `Speech Commands`: Based on the dataset released by Warden (2018). Readers recite one of 35 words. The task is then to classify which of the 35 words was spoken from a $16kHz$ one-dimensional audio recording. There are 35 different classes, each representing one of the words in the vocabulary. Sequences are all of the same length ($16,000$). There are $24,482$ training examples, $5,246$ validation examples, and $5,247$ test examples. Data is normalized to be zero mean and have a standard deviation of $0.2$.

- `Speech Commands × 0.5`: Temporally sub-sampled version of `Speech Commands`, where the validation and test datasets only are sub-sampled by a factor of $1/0.5$, and are therefore shortened to length $8,000$. No subsequent padding is applied. The training dataset is not subsampled.

- `Sequential MNIST`: (sMNIST) 10-way digit classification from a $28 \times 28$ grayscale image of a handwritten digit, where the input image is flattened into a $784$-length scalar sequence.

- `Permuted Sequential MNIST`: (psMNIST) 10-way digit classification from a $28 \times 28$ grayscale image of a handwritten digit, where the input image is flattened into a $784$-length scalar sequence. This sequence is then permuted using a fixed order.

- `Sequential CIFAR`: (sCIFAR): 10-way image classification using the CIFAR-10 dataset. Identical to `image`, except that full colour images are input as a $1,024$-length input sequence, where each input is an (R,G,B) triple.

- `Pendulum Regression`: Reproduced from Becker et al. (2019) and Schirmer et al. (2022). The input sequence is a $24 \times 24$ grayscale rendering of a pendulum, driven by a random torque process. The images pixels are corrupted by a noise process that is correlated in time. The pendulum is simulated for 100 timesteps, and 50 frames are irregularly sampled without replacement from the simulation. The objective is to estimate the sine and cosine of the angle of the pendulum. A train/validation/test split of $2,000/1,000/1,000$ is used.

## H  BACKGROUND ON PARALLEL SCANS FOR LINEAR RECURRENCES

For the interested reader, this section provides more background on using a parallel scan for a linear recurrence, as well as a simple example to illustrate how it can compute the recurrence in parallel. The parallelization of scan operations has been well studied (Ladner & Fischer, 1980; Lakshmivarahan & Dhall, 1994; Blelloch, 1990), and many standard scientific computing libraries contain efficient implementations. We note the linear recurrence we consider here is a specific instance of the more general setting discussed in Section 1.4 of Blelloch (1990).

Computing a general parallel scan requires defining two objects:

- The initial elements the scan will operate on.

- A binary associative operator • used to combine the elements.

To compute a length $L$ linear recurrence, $x_k = \overline{\mathbf{A}} x_{k-1} + \overline{\mathbf{B}} u_k$, we will define the $L$ initial elements, $c_{1:L}$, such that each element $c_k$ is the tuple

$$c_k = (c_{k,a}, c_{k,b}) := (\overline{\mathbf{A}}, \ \overline{\mathbf{B}} u_k). \tag{33}$$

These $c_{1:L}$ will be precomputed prior to the scan. Having created the list of elements for the scan to operate on, we define the binary operator • for the scan to use on this linear recurrence as

$$q_i \bullet q_j := \left(q_{j,a} \odot q_{i,a}, \ q_{j,a} \otimes q_{i,b} + q_{j,b}\right), \tag{34}$$

where $q_k$ denotes an input element to the operator that could be the initial elements $c_k$ or some intermediate result, $\odot$ denotes matrix-matrix multiplication, $\otimes$ denotes matrix-vector multiplication and $+$ denotes elementwise addition. We show that this operator is associative at the end of this section.

**Simple example using binary operator**  We can illustrate how • can be used to compute a linear recurrence in parallel with a simple example. Consider the system $x_k = \overline{\mathbf{A}} x_{k-1} + \overline{\mathbf{B}} u_k$, and a length $L = 4$ sequence of inputs $u_{1:4}$. Assuming $x_0 = 0$, the desired latent states from this recurrence are:

$$x_1 = \overline{\mathbf{B}} u_1 \tag{35}$$

$$x_2 = \overline{\mathbf{A}}\,\overline{\mathbf{B}} u_1 + \overline{\mathbf{B}} u_2 \tag{36}$$

$$x_3 = \overline{\mathbf{A}}^2 \overline{\mathbf{B}} u_1 + \overline{\mathbf{A}}\,\overline{\mathbf{B}} u_2 + \overline{\mathbf{B}} u_3 \tag{37}$$

$$x_4 = \overline{\mathbf{A}}^3 \overline{\mathbf{B}} u_1 + \overline{\mathbf{A}}^2 \overline{\mathbf{B}} u_2 + \overline{\mathbf{A}}\,\overline{\mathbf{B}} u_3 + \overline{\mathbf{B}} u_4 \tag{38}$$

We first note that • can be used to compute this recurrence sequentially. We can initialize the scan elements $c_{1:4}$ as in (33), and then sequentially scan over these elements to compute the output elements $s_i = s_{i-1} \bullet c_i$. Defining $s_0 := (\mathbf{I}, 0)$ where $\mathbf{I}$ is the identity matrix, we have for our example:

$$s_1 = s_0 \bullet c_1 = (\mathbf{I}, \ 0) \bullet (\overline{\mathbf{A}}, \ \overline{\mathbf{B}} u_1) = (\overline{\mathbf{A}} \mathbf{I}, \ \overline{\mathbf{A}} 0 + \overline{\mathbf{B}} u_1) = (\overline{\mathbf{A}}, \ \overline{\mathbf{B}} u_1) \tag{39}$$

$$s_2 = s_1 \bullet c_2 = (\overline{\mathbf{A}}, \ \overline{\mathbf{B}} u_1) \bullet (\overline{\mathbf{A}}, \ \overline{\mathbf{B}} u_2) = (\overline{\mathbf{A}}^2, \ \overline{\mathbf{A}}\,\overline{\mathbf{B}} u_1 + \overline{\mathbf{B}} u_2) \tag{40}$$

$$s_3 = s_2 \bullet c_3 = (\overline{\mathbf{A}}^2, \ \overline{\mathbf{A}}\,\overline{\mathbf{B}} u_1 + \overline{\mathbf{B}} u_2) \bullet (\overline{\mathbf{A}}, \ \overline{\mathbf{B}} u_3) = (\overline{\mathbf{A}}^3, \ \overline{\mathbf{A}}^2 \overline{\mathbf{B}} u_1 + \overline{\mathbf{A}}\,\overline{\mathbf{B}} u_2 + \overline{\mathbf{B}} u_3) \tag{41}$$

$$s_4 = s_3 \bullet c_4 = (\overline{\mathbf{A}}^3, \ \overline{\mathbf{A}}^2 \overline{\mathbf{B}} u_1 + \overline{\mathbf{A}}\,\overline{\mathbf{B}} u_2 + \overline{\mathbf{B}} u_3) \bullet (\overline{\mathbf{A}}, \ \overline{\mathbf{B}} u_4) \tag{42}$$

$$= (\overline{\mathbf{A}}^4, \ \overline{\mathbf{A}}^3 \overline{\mathbf{B}} u_1 + \overline{\mathbf{A}}^2 \overline{\mathbf{B}} u_2 + \overline{\mathbf{A}}\,\overline{\mathbf{B}} u_3 + \overline{\mathbf{B}} u_4). \tag{43}$$

Note that the second entry of each of the output tuples, $s_{i,b}$, contains the desired $x_i$ computed above. Computing the scan in this way requires four sequential steps since each $s_i$ depends on $s_{i-1}$.

Now consider how we can use this binary operator to compute the recurrence in parallel. We will label the output elements of the parallel scan as $r_{1:4}$ and define $r_0 = (\mathbf{I}, 0)$. We will first compute the even indexed elements $r_2$ and $r_4$, and then compute the odd indexed elements $r_1$ and $r_3$. We start by applying the binary operator • to adjacent pairs of our initial elements $c_{1:4}$ to compute $r_2$ and the

intermediate result $q_4$, and we then repeat this process to compute $r_4$ by applying $\bullet$ to $r_2$ and $q_4$:

$$r_2 = c_1 \bullet c_2 = (\overline{\mathbf{A}}, \ \overline{\mathbf{B}}u_1) \bullet (\overline{\mathbf{A}}, \ \overline{\mathbf{B}}u_2) = (\overline{\mathbf{A}}^2, \ \overline{\mathbf{A}}\overline{\mathbf{B}}u_1 + \overline{\mathbf{B}}u_2) \tag{44}$$

$$q_4 = c_3 \bullet c_4 = (\overline{\mathbf{A}}, \ \overline{\mathbf{B}}u_3) \bullet (\overline{\mathbf{A}}, \ \overline{\mathbf{B}}u_4) = (\overline{\mathbf{A}}^2, \ \overline{\mathbf{A}}\overline{\mathbf{B}}u_3 + \overline{\mathbf{B}}u_4) \tag{45}$$

$$r_4 = r_2 \bullet q_4 = (\overline{\mathbf{A}}^2, \ \overline{\mathbf{A}}\overline{\mathbf{B}}u_1 + \overline{\mathbf{B}}u_2) \bullet (\overline{\mathbf{A}}^2, \ \overline{\mathbf{A}}\overline{\mathbf{B}}u_3 + \overline{\mathbf{B}}u_4) \tag{46}$$

$$= (\overline{\mathbf{A}}^4, \ \overline{\mathbf{A}}^3\overline{\mathbf{B}}u_1 + \overline{\mathbf{A}}^2\overline{\mathbf{B}}u_2 + \overline{\mathbf{A}}\overline{\mathbf{B}}u_3 + \overline{\mathbf{B}}u_4). \tag{47}$$

Now we will compute the odd indexed elements $r_1$ and $r_3$, using the even indexed $r_0$ and $r_2$, as $r_k = r_{k-1} \bullet c_k$:

$$r_1 = r_0 \bullet c_1 = (I, \ 0) \bullet (\overline{\mathbf{A}}, \ \overline{\mathbf{B}}u_1) = (\overline{\mathbf{A}}, \ \overline{\mathbf{B}}u_1) \tag{48}$$

$$r_3 = r_2 \bullet c_3 = (\overline{\mathbf{A}}^2, \ \overline{\mathbf{A}}\overline{\mathbf{B}}u_1 + \overline{\mathbf{B}}u_2) \bullet (\overline{\mathbf{A}}, \ \overline{\mathbf{B}}u_3) = (\overline{\mathbf{A}}^3, \ \overline{\mathbf{A}}^2\overline{\mathbf{B}}u_1 + \overline{\mathbf{A}}\overline{\mathbf{B}}u_2 + \overline{\mathbf{B}}u_3). \tag{49}$$

Note that the second entry of each of the output tuples, $r_{k,b}$, corresponds to the desired $x_k$. Inspecting the required dependencies for each application of $\bullet$, we see that $r_2$ and the intermediate result $q_4$ can be computed in parallel. Once $r_2$ and $q_4$ are computed, $r_1$, $r_3$ and $r_4$ can all be computed in parallel. We have therefore reduced the number of sequential steps required from four in the sequential scan version to two in the parallel scan version. This reduction in sequential steps becomes important when the sequence length is large since, given sufficient processors, the parallel time scales logarithmically with the sequence length.

**Associativity of binary operator** Finally, for completeness, we show that the binary operator $\bullet$ is associative:

$$(q_i \bullet q_j) \bullet q_k = (q_{j,a} \odot q_{i,a}, \ q_{j,a} \otimes q_{i,b} + q_{j,b}) \bullet q_k \tag{50}$$

$$= (q_{k,a} \odot (q_{j,a} \odot q_{i,a}), \ q_{k,a} \otimes (q_{j,a} \otimes q_{i,b} + q_{j,b}) + q_{k,b}) \tag{51}$$

$$= ((q_{k,a} \odot q_{j,a}) \odot q_{i,a}, \ q_{k,a} \otimes (q_{j,a} \otimes q_{i,b}) + q_{k,a} \otimes q_{j,b} + q_{k,b}) \tag{52}$$

$$= ((q_{k,a} \odot q_{j,a}) \odot q_{i,a}, \ (q_{k,a} \odot q_{j,a}) \otimes q_{i,b} + q_{k,a} \otimes q_{j,b} + q_{k,b}) \tag{53}$$

$$= q_i \bullet (q_{k,a} \odot q_{j,a}, \ q_{k,a} \otimes q_{j,b} + q_{k,b}) \tag{54}$$

$$= q_i \bullet (q_j \bullet q_k) \tag{55}$$

