# OpenReview forum: "Simplified State Space Layers for Sequence Modeling"
_ICLR.cc/2023/Conference — ICLR 2023 notable top 5%_

### Official Review · Reviewer_Gzd2 · 2022-10-23

**Confidence:** 3
**Correctness:** 4
**Technical Novelty And Significance:** 3
**Empirical Novelty And Significance:** 3
**Recommendation:** 8

**Clarity, Quality, Novelty And Reproducibility:**

The paper is very clear and high quality. It is easy to be reproduced with code given in the appendix, and linked (although I was not able to access it). One of the advantages of the approach is also that it is much easier to implement and use than the S4 layers as it does not require both the sequential and convolution & FFT mode implementation (only sequential).
While the model itself "just" extends previous work, there are several technical contributions that are novel in this context.


**Strength And Weaknesses:**

This paper is very well written, precise, and easy to follow. I found the background on S4 layers incl. the diagram very useful and more accessible than the original paper, and I also think the authors made a good choice in explaining their extension S5 by contrasting them to S4. It is also great to see the easy-to-implement jax code in the appendix.

The experimental evaluation is extensive, with many and the most important baselines, evaluated on commonly used long-range-dependency benchmarks. It is especially nice to see ablation studies in Appendix E, evaluating different model sizes (making it more or less similar to S4) as well as initialisation and continuous-time (vs. direct discrete-time implementation), showing that these are the most important features.

I did not find major problems or concerns.
But I would be interested if the relation between S4 and S5 (App. D) could be made with less restrictions. Especially assumption 2 seems a bit strict?
I would also have liked to see how the model deals with uncertainty/unpredictability, which is often the case in forecasting (e.g. the stock market would be an extreme case). The sequential image classification tasks are a bit unrealistic/artificial. But I do realise this is not the goal of the paper, and previous work from this domain also does not evaluate in such settings.

**Summary Of The Paper:**

This paper presents an extension of structured state space sequence (S4) layers [Gu et al. 2021], which the authors refer to as S5 layers.
While S4 layers use a bank of (deterministic) linear continuous state-space models with subsequent mixing layers, S5 uses a single multi-input multi-output SSM that does not require mixing layers. Furthermore, S4 parallelises the (offline) computation for the sequence of linear operations using a 1D convolution identity and an implementation in frequency domain (FFT). This has been done for computational reasons (parallelisation). However, this paper shows that S5 layers can match the computational and memory complexity (also shown empirically), by using the scan operation.
The authors show connections between S4 and S5, which they use for building on the initialization of S4. The authors also show via extensive ablation studies (in the supplementary material), that the initialisation as well as the continuous-time formulation with learned discretisation timescale parameters are the main drivers for these models.


**Summary Of The Review:**

Overall I enjoyed reading the paper, it is high-quality work, in terms of method, experimentation, and the presentation.
I did not find any major problems or concerns.

---

> ### Author Response · Authors · 2022-11-10
> **Gzd2 Response**
>
> Thank you again for your positive feedback.  We have added anonymized code in the supplement.
>
> *R.e. Assumptions in Section 4*: The derivation in Section 4 was used to establish _a_ connection between a restricted version of S4 and a restricted version of S5 to help explore certain architectural, initialization and parameterization choices.  We agree that in the most general case the tied state matrix assumption could be restrictive.  The original S4 paper and the more recent S4D paper did however report that strong performance can be achieved with tied state matrices (cf. Assumption 2). We verify in Rows 3 and 4 of Appendix E/Table 5 that the restricted version of S5 still performs comparably to the best (unrestricted) S4 models on most of the LRA tasks. We clarify however, _we do not enforce any restrictions when training S5 models in practice_.  Nonetheless, we agree that further exploring the theoretical relationship and differences between S4 and S5 are interesting questions.
>
> *R.e. Uncertainty/unpredictability*: This is a great point.  We agree forecasting in uncertain environments is an interesting application. While one could straightforwardly use S5 to parameterize a stochastic emission (while retaining deterministic internal dynamics), we are also interested in future directions of S5 that involve stochastic SSM dynamics. Part of our initial motivation for S5 was to remove some of S4’s machinery to allow for closer connections to probabilistic methods (e.g. parallel Kalman filtering/smoothing). We will include discussion of these points in any final version.
>
> Thank you again for your feedback.  We look forward to answering any further questions you have!
>
> – The S5 Authors.

---

> > ### Author Response · Authors · 2022-11-18
> > **Final Thoughts**
> >
> > Hi Reviewer Gzd2,
> >
> > As the rebuttal period is closing shortly, please let us know if you have any further questions or if we can provide further clarification!
> >
> > --The S5 Authors

---

> > > ### Comment · Reviewer_Gzd2 · 2022-11-18
> > > **Response to Authors**
> > >
> > > My questions have been addressed, thank you.
> > > I am also looking forward to read the discussion about connections to probabilistic SSMs with Kalmen filtering/smoothing for inference.

---

### Official Review · Reviewer_w99z · 2022-10-24

**Confidence:** 4
**Correctness:** 4
**Technical Novelty And Significance:** 3
**Empirical Novelty And Significance:** Not applicable
**Recommendation:** 8

**Clarity, Quality, Novelty And Reproducibility:**

The paper is clearly written and the quality of the presented content meets the conference standards.
The suggested implementation is novel in a way that it improves the S4 layer by a more efficient computation method. It would have been nice, though, if the authors would have provided the corresponding implementations already via OpenReview.

**Strength And Weaknesses:**

 - [+] The paper is very well written with a concise and clearly structured background section.
 - [+] All experiments are repeated multiple times with different seeds, leading to more stable and thus scientifically valuable results.
 - [+] Detailed and comprehensive comparison with S4 and its components with helpful illustrations and descriptions.
 - [+] The reviewer highly acknowledges the inclusion of concurrent work in the comparison.
 - [-] Although all experiments are repeated multiple times, no error bars are reported. Especially in cases where multiple methods perform similarly, a corresponding deviation measure proves to be really
       helpful for judging.
 - [-] It is not clear what the reported scores in Table 1 are. Assuming the scores to be accuracy scores, the average accuracy provides only limited insight under the assumption that the tasks are of
       varying difficulty (i.e. different number of classes with different relative class sizes, etc.). The reviewer suggests reporting the average rank for each method instead.
 - [-] Regarding the results reported in Table 3: are the results marked with an asterisk (*) computed on 5 seeds, and all others (i.e. „CRU (our run)“ and „S5“) on 20 seeds? If yes, this might skew the
       result. The reviewer suggests using the same test protocol for all methods in this case. If not, the reviewer suggests clarifying the use of different seeds.



**Summary Of The Paper:**

The paper introduces the new state space layer S5 which combines multiple single-input single-output state space models to a single multiple-input multiple-output state space model. The proposed state space layer can be efficiently implemented by a parallel scan. S5 achieves state-of-the-art results on multiple benchmarks, including the difficult Path-X task.

**Summary Of The Review:**

The paper is very well written and easy to follow. All necessary background information is concisely introduced. The authors provide extensive analyses w.r.t. ablations, runtime and memory footprints. Some ambiguities and questions still remain, whereas the reviewer hopes they will be clarified during the rebuttal period.
*The reviewer will likely increase the score if the weaknesses and questions are clarified in sufficient detail.*

### Questions

 - What might be the main reason or intuition why S5 has better results than S4 w.r.t. scores, besides computational efficiency/runtime?
 - Why wasn’t S4 used as a baseline in Table 3?
 - What is the hyperparameter search space of the baseline methods?

---
### Update

The response of the authors addressed our concerns. Therefore, the score has been raised (as promised).

---

> ### Author Response · Authors · 2022-11-10
> **w99z Response**
>
> Thank you again for your positive feedback.  We have added anonymized code in the supplement.
>
> *R.e. Deviation Measures*:  We omitted deviation measures from Table 1 in the main text for clarity of presentation (as there were no great “surprises” in the deviations of S5 compared to the S4 methods).  Deviation measures were however included in the expanded LRA table in Appendix F.1 / Table 7.  We will add a note that deviations are included in the supplement.  We will include a supplementary table with deviation measures for SC35 for a final version (the deviation measures were again similar for both S4 and S5).  Deviation measures were included in the pendulum results tables in the main text and appendix.
>
> *R.e. Table 1 Scores*:  You are correct, Table 1 shows the accuracy of each method/task.  We have added clarification on this.  Showing the individual task and average scores for LRA, with bolding and underlining for first and second places methods has precedent:  see the original LRA  [paper](https://arxiv.org/abs/2011.04006), table on the LRA [GitHub](https://github.com/google-research/long-range-arena), the original S4 [paper](https://arxiv.org/pdf/2111.00396.pdf), as well as the concurrent work [MEGA](https://arxiv.org/abs/2209.10655). We followed this convention in our  submission. However, we like your suggestion and will also include an average rank column.
>
> *R.e. Pendulum reporting*:  You are correct, the results marked with an asterisk were run with five seeds, and are reported directly as in Schirmer et al. [2022].  We ran twenty seeds to provide as reliable a baseline as possible for the best-performing models.  If the reviewer would prefer, we can report in the main text the result based on five seeds for consistency.  The results are very similar, with S5 having an MSE of 3.45e-3 $\pm$ 0.159e-3 and CRU having an MSE of 3.87e-3 $\pm$ 0.170e-3 (across five seeds).
>
> *R.e. Questions*:
>
> (1)  A possible reason for the improved performance of S5 is that the MIMO SSM allows different dynamics and couplings of the layer input features, compared to the independent SISO SSMs that S4 uses to process each channel of the input features (that are then coupled by the mixing layer).  Rows 3 and 4 of Appendix E.1/Table 5 represent the performance of S5 in a setting where the dynamics of the S5 SSM have been constrained to be as similar as possible to the dynamics of the S4 models (per the discussion in Section 4.1). In this case, the _restricted_ S5 model performs comparably to S4.  Though we have performed a number of studies with ablations and constraints, the relationship and performance differences between S4 methods and S5 remains an interesting topic for further investigation.
>
> (2)  On why S4 was not used as a baseline for the pendulum regression task: As mentioned in Section 3.3, the efficient convolution mode of S4 requires a time-invariant system which prevents its use to efficiently handle irregularly sampled data. This is because performing a different discretization at each timestep to account for the irregular spacings results in a time-varying system. Thus, S4 would have to be run in its fully sequential recurrent mode to naturally handle irregularly sampled data, defeating one of the main purposes of using state space layers (being able to parallelize across time in offline settings) rather than the RNN baselines. S5’s parallel scan on the other hand allows for naturally handling time-varying systems and irregularly sampled data efficiently.
>
> (3)  The baseline methods’ performance were taken from the original papers.  For our run of CRU, we used the optimal hyperparameters reported in the original paper.  We have included more explicit information on this.
>
> Thank you again for your feedback.  We look forward to answering any further questions you have!
>
> – The S5 Authors.

---

> > ### Author Response · Authors · 2022-11-18
> > **Final Thoughts**
> >
> > Hi Reviewer w99z,
> >
> > As the rebuttal period is closing shortly, please let us know if you have any further questions or if we can provide further clarification!
> >
> > --The S5 Authors

---

### Official Review · Reviewer_2DkJ · 2022-10-26

**Confidence:** 3
**Clarity, Quality, Novelty And Reproducibility:** the paper is well-motivated and well-…
**Correctness:** 3
**Technical Novelty And Significance:** 3
**Empirical Novelty And Significance:** 3
**Recommendation:** 8

**Strength And Weaknesses:**

The proposed layer, s5, replaces the frequency-domain approach used by S4 with a recurrent, time-domain approach bypassing non-trivial steps needed for S4. The method achieves high performance with comparable complexity of s4.

The source code link provided in the paper isn't working, The authors should publish anonymized code for the review purposes with the correct link.

**Summary Of The Paper:**

The paper addresses the problem of long-range sequence modeling. The authors propose the s5 layer, which is built atop and modified version of structured state-space-sequence, s4. Utilizing parallel scans, s5 improves over s4 in terms of performance and complexity. They show improved results on LRA benchmark without incurring additional complexity comparing with S4.



**Summary Of The Review:**

This is a well-written, well-motivated paper.

---

> ### Author Response · Authors · 2022-11-10
> **2DkJ Response**
>
> Thank you again for your positive feedback.  We have added anonymized code in the supplement.  We look forward to answering any further questions you have!
>
> – The S5 Authors.

---

> > ### Author Response · Authors · 2022-11-18
> > **Final Thoughts**
> >
> > Hi Reviewer 2Dkj,
> >
> > As the rebuttal period is closing shortly, please let us know if you have any further questions or if we can provide further clarification!
> >
> > --The S5 Authors

---

### Author Response · Authors · 2022-11-10
**General Response**

We thank the reviewers for taking the time to review our submission and for their positive feedback.  We presented the S5 layer, a sequence modeling layer building on the popular S4 layer.  We were especially happy that the reviewers valued the presentation of both existing and new material, and that they all saw the benefits of S5.

The common critique from each of the reviewers was that we did not include our code in the initial submission.  We apologize for the confusion: the link to our Github repository was redacted to preserve anonymity.   We have updated the supplementary materials to include an anonymized version of the documented core code.

We appreciate that the reviewers truly engaged with our submission and critically assessed the work.  We will now respond to individual reviewers in more detail.

– The S5 Authors.

---

### Decision · Program_Chairs · 2023-01-20

**Decision:**

Accept: notable-top-5%

**Justification For Why Not Higher Score:**

N/A

**Justification For Why Not Lower Score:**

A promising advance in long-range sequence modeling with deep neural nets coming from a very different perspective than self-attention and transformers, but which brings it on par with those much more widely researched methods. In my opinion, this would be of widespread interest at ICLR and would make a good choice for oral presentation.

**Metareview: Summary, Strengths And Weaknesses:**

For handling long sequential inputs as required in many modern applications of neural nets, this paper proposes an "S5" layer, "Simplified State Space Layers for Sequential Modeling", which is more expressive than the previous "S4" layer from ICLR 2021, while retaining computational efficiency. The two key changes: (1) single-input single-output banks + mixing layers for dealing with multi-dimensional inputs are changed to multi-input multi-output processing. (2) time-domain processing rather than frequency domain.

Strengths:
- The technical improvements over S4 are important, lifting restrictions of the earlier, already quite successful model.
- The results on various long-range sequence modeling tasks show comparable performance with the very best transformer attention based models, without incorporating any of that machinery.
- Implementation appears straightforward, suggesting ease of use by others in the community.
- Evaluated against the very latest work possible, including even work released during the week of the ICLR deadline.
- Well-presented.

Weaknesses:
- Evaluations are on somewhat synthetic tasks, although this is in keeping with the current practice in long-range sequence modeling.
- It would be good to demonstrate some qualitative examples, such as visualizing how information from various stages of historical input are used in producing current outputs.

**Note From Pc:**

if the above contains the word "oral" or "spotlight" please see: "oral" presentation means -> notable-top-5% and "spotlight" means -> notable-top-25%. As stated in our emails, we are disassociating presentation type from AC recommendations